# CAJAL enables analysis and integration of single-cell morphological data using metric geometry

Kiya W. Govek[1], Patrick Nicodemus[1], Yuxuan Lin[2], Jake Crawford[3], Artur B. Saturnino [2], Hannah Cui[2], Kristi Zoga[1], Michael P. Hart [1] & Pablo G. Camara [1,4,5]

High-resolution imaging has revolutionized the study of single cells in their spatial context. However, summarizing the great diversity of complex cell shapes found in tissues and inferring associations with other single-cell data remains a challenge. Here, we present CAJAL, a general computational framework for the analysis and integration of single-cell morphological data. By building upon metric geometry, CAJAL infers cell morphology latent spaces where distances between points indicate the amount of physical deformation required to change the morphology of one cell into that of another. We show that cell morphology spaces facilitate the integration of single-cell morphological data across technologies and the inference of relations with other data, such as single-cell transcriptomic data. We demonstrate the utility of CAJAL with several morphological datasets of neurons and glia and identify genes associated with neuronal plasticity in *C. elegans*. Our approach provides an effective strategy for integrating cell morphology data into single-cell omics analyses.

Since the advent of staining techniques in the nineteenth century, cell morphology has become one of the most described phenotypes in biology. The idea that the morphology of a cell is related to its function has been central to major discoveries, such as the neuron doctrine[1], the molecular basis of sickle cell disease[2], and the pathways for cell migration and chemotropic sensing[3]. In the nervous system, whole-cell patch-clamp has enabled the morphological reconstruction and electrophysiological recording of >100,000 neurons[4]. The incorporation of high-throughput single-cell RNA-seq onto patch-clamp, known as Patch-seq[5–9], has opened the door to deeper characterizations that include morphological, transcriptomic, and electrophysiological information from the same cells[10]. More broadly, the recent explosion of spatially resolved technologies for single-cell transcriptomics has

transformed the study of cells in their spatial context by enabling researchers to infer links between morphological and molecular phenotypes[11–13]. The potential of this new array of techniques is immense, not only for cell taxonomic purposes, but also for uncovering the pathways that are associated with, and may ultimately drive, the morphological diversification and plasticity of cells.

Algorithms for cell morphometry seek to determine similarities among the morphology of individual cells in digitally reconstructed microscopy images. Current methods extract a set of shape descriptors that summarize the morphology of each cell. Simple geometric descriptors like the area and perimeter of the cell[14,15] can be applied to most cell types, but have limited power to accurately discriminate complex cell morphologies like those of neurons and glia. On the other

[1]Department of Genetics, Perelman School of Medicine, University of Pennsylvania, Philadelphia, PA 19104, USA. [2]Department of Mathematics, School of Arts and Sciences, University of Pennsylvania, Philadelphia, PA 19104, USA. [3]Genomics and Computational Biology Graduate Group, Perelman School of Medicine, University of Pennsylvania, Philadelphia, PA 19104, USA. [4]Institute for Biomedical Informatics, Perelman School of Medicine, University of Pennsylvania, Philadelphia, PA 19104, USA. [5]Center for Artificial Intelligence and Data Science for Integrated Diagnostics, Perelman School of Medicine, University of Pennsylvania, Philadelphia, PA 19104, USA. ✉e-mail: pcamara@pennmedicine.upenn.edu

hand, more complex cell-type-specific descriptors, such as neuronal branching patterns[16–18], need to be tailored to the specific cell type of interest and cannot be applied broadly. In addition, they are arbitrary with respect to the features that are used, and the weight assigned to them. To overcome these limitations, other methods introduce similarity scores based on tree alignment[19,20] or decomposition in Fourier, Zernike, or spherical harmonic moments[21–24]. However, these methods require building combinations of descriptors that are invariant under rigid transformations or carefully pre-aligning the cells using Procrustes analysis, and they fail to quantify morphological differences between highly dissimilar cells. In general, none of the current approaches reflect the biophysical processes involved in cell morphological changes or lead to an actual distance function in the cell morphology space. These limitations have precluded the development of advanced algebraic and statistical approaches for the analysis of cell morphology data, such as methods for integrating these data, constructing consensus cell morphologies, or inferring cell state trajectories associated with morphological processes.

Here, we build upon recent developments in applied metric geometry and shape registration[25–27] to establish a general computational framework for studying complex and heterogeneous cell morphologies across the broad range of cells found in tissues. This framework enables the characterization of morphological cellular processes from a biophysical perspective and produces an actual mathematical distance upon which rigorous algebraic and statistical analytic approaches can be built. Our analyses show that this approach has the generality and stability of simple geometric shape descriptors, the discriminative power of cell-type-specific descriptors, and the unbiasedness and hierarchical structure of moments-based descriptors. Using this framework, we address several outstanding methodological roadblocks in relating cell morphology to molecular content and function, including the integration of morphological information across experiments and technologies and the combined analysis of morphological, molecular, and physiological information of individual cells. Applying this framework to Patch-seq, patch-clamp, fluorescence micro-optical sectioning tomography (fMOST), electron, and two-photon microscopy data of neurons and glia, we identify some of the genetic and molecular programs that are associated with the plasticity of neurons. Taken together, the results of these analyses demonstrate that the application of metric geometry to the study of cell morphology not only increases the accuracy and versatility of cell morphology analyses, but also enables currently unavailable analyses such as the integration of cell morphological data across experiments. We have implemented this analytic framework in an open-source software[28], called CAJAL, which we expect will be useful to researchers working with single-cell morphological data of any kind.

## Results

### A general framework for the quantitative analysis of cell morphology data

In its simplest formulation, the study of cell morphology involves the quantitative comparison of cell shapes irrespective of distance-preserving transformations (*isometries*), such as rotations and translations. From a mathematical standpoint, this is a problem of metric geometry. The Gromov–Hausdorf (GH) distance measures how far two *compact metric spaces* are from being isometric[29,30]. In physical terms, it determines the minimum amount of deformation required to convert the shape of an object into that of another. The use of the GH distance to describe cell shapes is therefore broadly applicable to any cell type, as it does not rely on geometric features that are particular to the cell type or require pre-aligning the cells to a reference shape. Because of these reasons, we sought to develop a general framework for cell morphometry by building upon these concepts in metric geometry.

Since computing the GH distance is intractable even for relatively small datasets, we based our approach upon a computationally efficient approximation, referred to as the Gromov–Wasserstein (GW) distance[25–27]. The GW distance preserves most of the mathematical properties of the GH distance and leads to an actual distance function[26]. Although its running time grows cubically with the number of points, its efficiency can be further improved by means of nearly linear-time approximations that build upon optimal transport regularization[31,32] and nesting strategies[33].

The starting point to our analytic framework is the 2D segmentation masks or 3D digital reconstructions of individual cells, which are discretized by evenly sampling points from their outline (Fig. 1a). For each cell, we compute the pairwise distance matrix ($d_i$) between its sampled points. Then, for each pair of cells, $i$ and $j$, we compute the GW distance between the matrices $d_i$ and $d_j$ using optimal transport (Fig. 1b). The result is a pairwise GW distance that quantifies the morphological differences between each pair of cells.

Different metrics for measuring distances between sampled points lead to different properties of the GW distance that may be advantageous in specific applications (Fig. 1a). For example, using Euclidean distance results in a GW matrix that accounts for the positioning of cell appendages, which can be particularly useful in the study of neuronal projections. On the other hand, using geodesic distance results in a GW matrix that is invariant under bending deformations of the cell, and it is therefore particularly sensitive to topological features such as the branching structure of cell appendages.

In all cases, the resulting GW distance can be thought of as a distance in a latent space of cell morphologies (Fig. 1c). In this latent space, each cell is represented by a point, and distances between cells indicate the amount of physical deformation needed to change the morphology of one cell into that of another. By formulating the problem in this way, we can use statistical and machine learning approaches to define cell populations based on their morphology; dimensionally reduce and visualize cell morphology spaces; integrate cell morphology spaces across tissues, technologies, and with other single-cell data modalities (for example, single-cell RNA-seq or ATAC-seq data); or infer trajectories associated with continuous morphological processes. For example, for datasets that contain both morphological and omics data, the Laplacian score for feature selection[34] can be used to identify omics features, such as gene expression or genetic variants, that are associated with differences in cell morphology. We have implemented these analyses in an open-source Python library, called CAJAL, which can be used with arbitrarily complex and heterogeneous cell populations[28] (Fig. 1d).

### GW cell morphology spaces accurately summarize complex cell shapes

To assess the ability of GW cell morphology spaces to summarize complex cell shapes, we applied CAJAL to the 3D basal and apical dendrite reconstructions of 506 neurons from the mouse visual cortex profiled with patch-clamp[35]. The resulting space of cell morphologies recapitulated the neuronal morphological types of the visual cortex (Fig. 2a). Cells with a similar morphology appeared in proximity in the UMAP representation of this space. Molecularly defined neuronal types were also localized in the representation (Fig. 2b), consistent with the presence of morphological characteristics that are unique to each molecular subtype. Excitatory and inhibitory neurons clustered separately, and individual neurons were organized in the cell morphology space according to their cortical layer and Cre driver line (Fig. 2b). Clustering the morphology space using Louvain community detection[36] partitioned it into 9 morphological populations. Using the metric structure of the cell morphology space, we then computed the medoid and average cell morphology for each cluster (Fig. 2c). These summaries accurately represented the main morphological

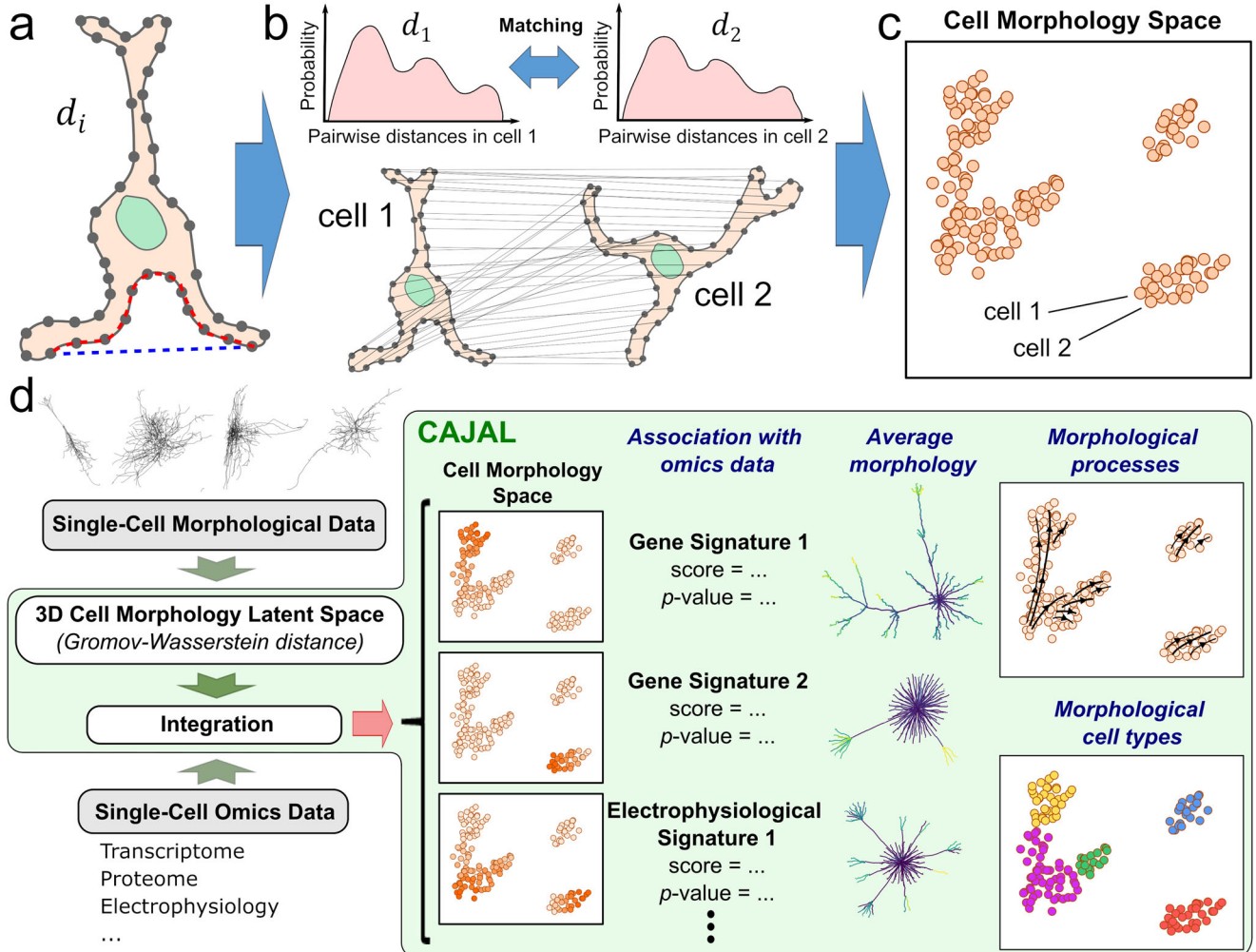

**Fig. 1 | A general framework for the quantitative analysis and integration of cell morphological data based on metric geometry. a** CAJAL takes as input the 2D or 3D cell segmentation masks or traces of a set of cells. For each cell, a set of points is evenly sampled from the outline and their pairwise distance matrix $d_i$ is computed. The Euclidean and geodesic distances between 2 sampled points is indicated with a blue and red dashed line, respectively. Different metrics for computing $d_i$ capture different aspects of cell morphology. **b** An optimal matching between the discretized morphologies of each pair of cells is established by computing the GW distance between their corresponding $d_i$ matrices. Computationally efficient approximations to the GW distance use optimal transport theory to establish a map between the distributions of sampled points pairwise distances in each cell. The value of the cost function at the minimum quantifies the amount of deformation that is needed to convert the shape of one cell in that of another. **c** The GW distance matrix can be thought of as a distance in a latent space of cell morphologies. This enables the application of statistical and machine learning methods for the analysis and integration of point clouds. **d** Overview of the open-source software CAJAL. The software takes single-cell morphological data in the form of segmentation masks or neuronal traces as input and enables its integration with other single-cell data modalities, the visualization and clustering of cell morphology spaces, the identification of molecular and electrophysiological features associated with changes in cell morphology, the computation of average or representative cell shapes, and the visualization of trajectories in the cell morphology space.

characteristics of the cell populations and were consistent with the diversity of neuronal morphologies found in the mouse visual cortex[35]. Specifically, known morphological types (m-types) of visual cortex neurons obtained by hierarchical clustering of pre-defined lists of morphological features[35] were localized in the cell morphology space (Supplementary Fig. 1).

Current methods for neuronal morphometry differ in their modeling assumptions, the use of predefined morphological features, and their capacity to produce a morphological distance function. To perform a quantitative evaluation of the ability of the GW distance to accurately summarize complex neuron morphologies in comparison to current neuronal morphometry methods, we analyzed four published datasets from the Allen Brain Institute and the BRAIN Initiative. These datasets comprised three different technologies for single-cell morphology profiling and included two Patch-seq datasets of the mouse visual[37] and motor cortices[38], a fMOST dataset of the mouse brain[39], and the patch-clamp dataset of the visual cortex[35] analyzed

above. For each of the four datasets, we assessed the ability of CAJAL and 6 other methods (Sholl analysis[16], L-measure[17], SNT[18], NBLAST[19], TMD[40], and ElasticP2P[41]) to identify morphological differences between molecularly defined neurons. In the case of the patch-clamp dataset, we considered neurons labeled with different Cre driver lines, for a total of 31 lines, with the understanding that each line preferentially labels distinct molecular neuronal types. In the case of the two Patch-seq datasets, we considered the classification of motor and visual cortex neurons into 9 and 6 known transcriptionally defined classes[37,38], respectively. In the case of the fMOST dataset, we considered the combination of Cre driver line and anatomic location, for a total of 8 neuronal types.

We first evaluated the performance of each method using the cell-type separation (CTS) score introduced in a previous comparative study of cell shape analysis methods[22]. This score quantifies the relative separation of molecularly and anatomically defined cell types in the morphology space produced by each method. Across all four

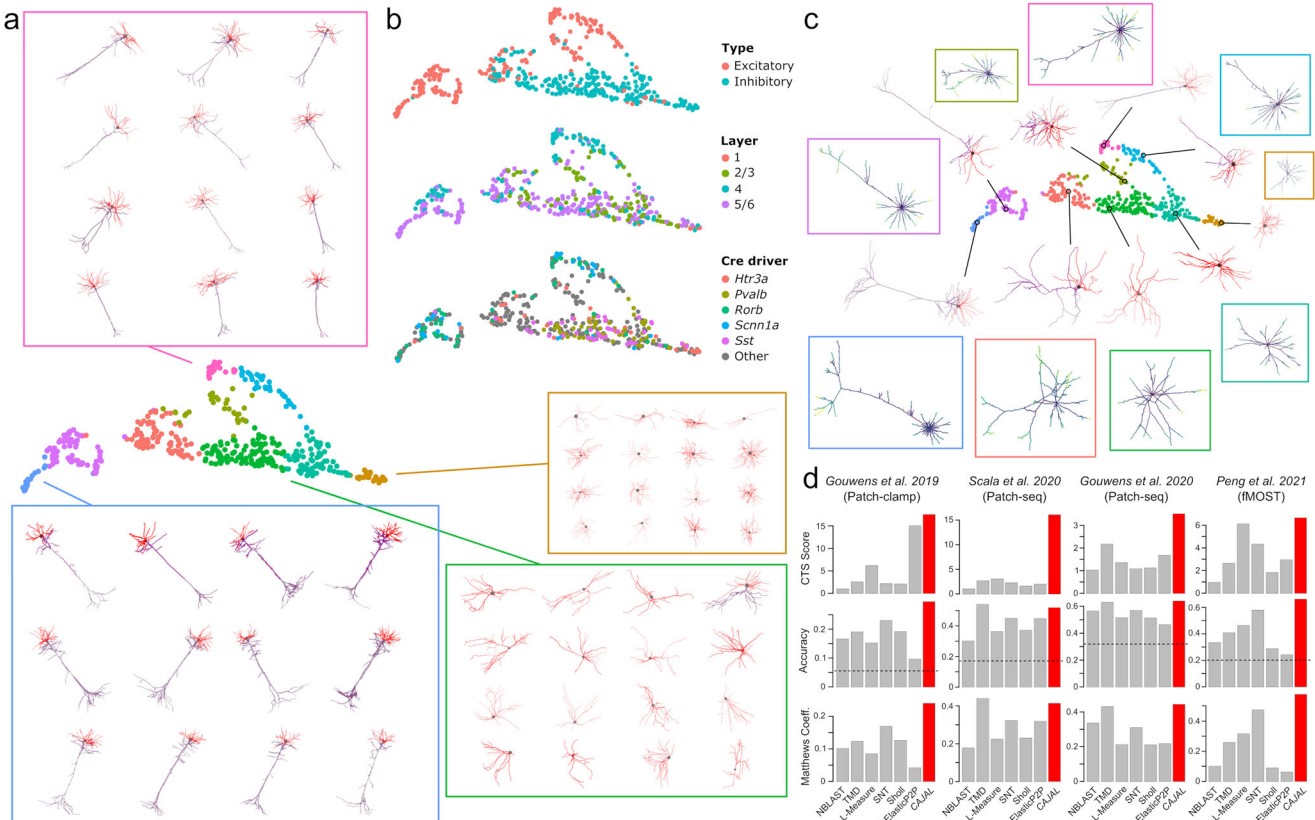

**Fig. 2 | Cell morphology spaces accurately summarize the complexity of cell shapes. a** UMAP representation of the cell morphology space of 506 neurons from the murine visual cortex profiled with whole-cell patch-clamp. The representation is colored by the morphological cell populations that resulted from clustering the cell morphology space using Louvain community detection. The morphologies of individual neurons randomly sampled from 4 of the populations are shown for reference. Apical and basal dendrites are indicated in purple and red, respectively. **b** The UMAP representation is colored by the neuronal type (top), cortical layer (middle), and Cre driver line (bottom). The GW cell morphology space captures morphological differences between neurons of different molecular type and anatomic location. **c** The metric structure of the GW morphology space enables performing algebraic operations such as averaging shapes. The figure shows the medoid (indicated with a circle) and average morphologies (in boxes) computed for each of the morphological populations in (**a**). **d** Cell-type separation (CTS) score, accuracy, and Matthews correlation coefficient (MCC) for CAJAL and six other neuronal morphometry methods in the task of predicting the molecular and/ or anatomic location of individual neurons in four datasets (one patch-clamp, two Patch-seq, and one fMOST datasets). The dashed line indicates the accuracy of a random classifier. More detailed information is presented in Supplementary Tables 1 and 2. Source data are provided as a Source data file.

datasets, CAJAL exhibited the highest CTS score among the 7 methods, with a score 3.1 times larger on average than that of the other methods (Fig. 2d and Supplementary Table 1). ElasticP2P and L-Measure achieved the second and third best scores, respectively, with values half the size of CAJAL's score.

We next trained a $k$ nearest neighbor classifier to predict the molecular type of each neuron based on its location in the cell morphology space. We estimated the accuracy of each method (percentage of correctly classified cells) using a cross-validation scheme (Methods). In this analysis, all methods are expected to have accuracies substantially smaller than 100%, since transcriptomic subtypes cannot be fully distinguished based on morphology. However, relative differences in accuracy in this analysis are indicative of differences in the ability of each method to capture meaningful biological information from cell morphology. We found that the accuracy of CAJAL varied between 29% and 66% depending on the dataset, and outperformed other methods in 22 out of the 24 dataset-method comparisons (Supplementary Table 2, Wilcoxon rank-sum test $p$ value $< 10^{-4}$). The second best-performing method was TMD, which produced similar results to CAJAL in the two Patch-seq datasets, achieving 99% and 106% accuracy with respect to CAJAL (TMD accuracy = 63%, 54%, *CAJAL* accuracy = 64%, 52%, respectively). However, the accuracy of TMD was substantially lower in the Patch-clamp and fMOST datasets, achieving respectively 64% and 62% accuracy with respect to CAJAL (TMD

accuracy = 19%, 41%, CAJAL accuracy = 29%, 66%, respectively). On average, the accuracy of CAJAL was 1.42 times higher than that of the other methods (Fig. 2d and Supplementary Table 2). SNT and TMD were the second and third most accurate methods, with an average accuracy 85.6% and 82.5% relative to CAJAL, respectively.

The use of accuracy as a metric for classification performance evaluation can offer inflated results in some circumstances[42]. For instance, in the above analysis the accuracy of a random classifier varied between 5% and 20% depending on the dataset (Supplementary Table 2). Matthews correlation coefficient (MCC) has been proposed as an alternative metric for classification performance evaluation that accounts for both true and false positives and negatives[42]. In our analysis, we found that the MCC of CAJAL was on average 1.84 times larger than that of the other methods, with the MCC of the second and third best performing methods (SNT and TMD) being on average 75.6% that of CAJAL (Fig. 2d and Supplementary Table 2).

To assess the stability of these results, we repeated these analyses for different choices of the intracellular pairwise distance function and sampling approach. We observed that using Euclidean distance to measure pairwise distances between sampled points offered more accurate predictions of the transcriptomic class of neurons than using geodesic distance (Supplementary Fig. 2a). This suggests that the relative position of neuronal appendages, and not only their topology, contains information about the transcriptomic class of the neuron.

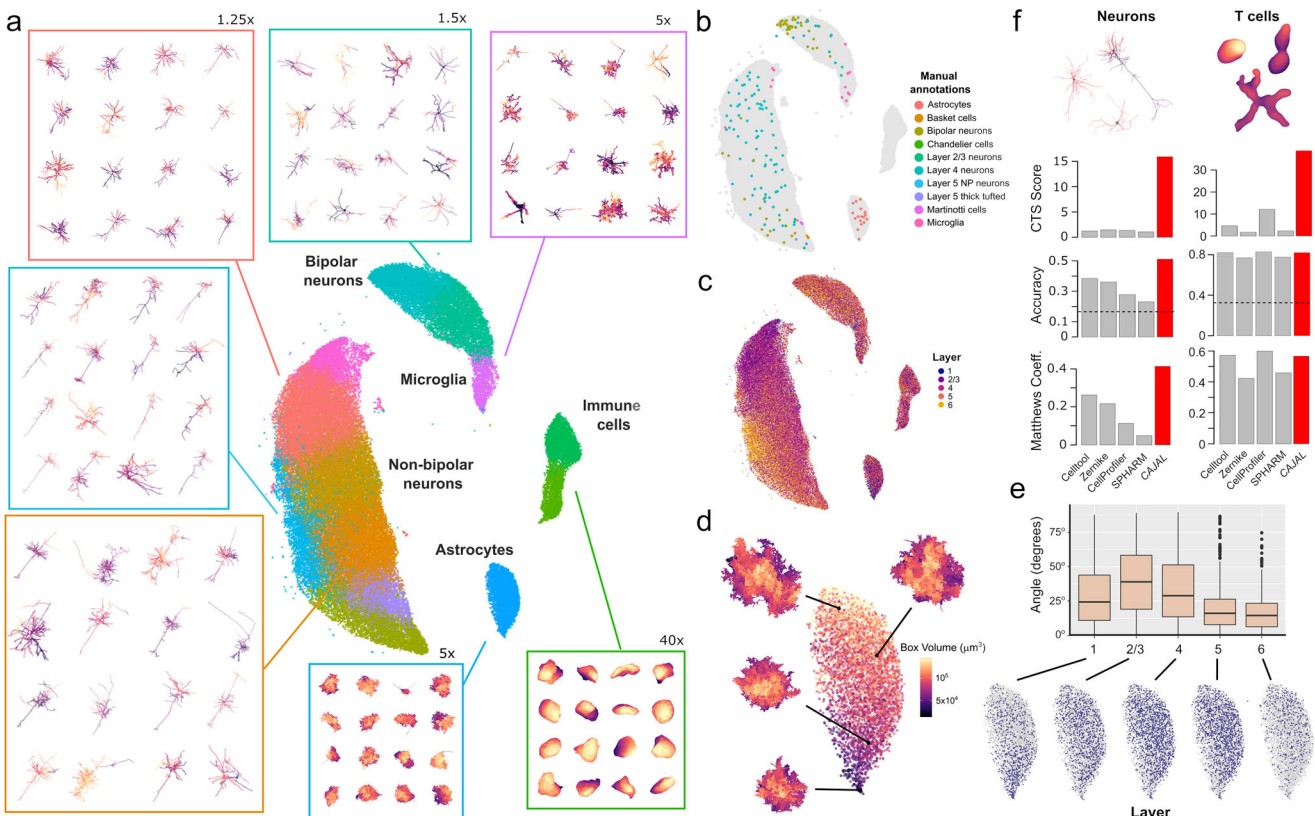

**Fig. 3 | Cell morphology spaces summarize cell shapes across heterogeneous cell types. a** UMAP representation of the cell morphology space of 70,510 cells from a cubic millimeter volume of the mouse visual cortex profiled by the MICrONS program using two-photon microscopy, microtomography, and serial electron microscopy[43]. The representation is colored by the cell populations identified by clustering of the cell morphology space. The morphology of randomly sampled cells from several populations is shown for reference. The magnification is indicated in cases where the morphology of the cells has been zoomed in to facilitate visualization. **b** The position of 185 cells that were manually annotated by the MICrONS program is indicated in the UMAP representation, showing consistency with the structure of the cell morphology space. **c** The UMAP representation of the cell morphology space is colored by the cortical layer of each cell. The morphology space recapitulates morphological differences between neurons and astrocytes from different cortical layers. **d** The part of the UMAP representation corresponding to astrocytes is colored by the volume of the minimum-size box containing the astrocyte. Astrocytes in the lower part of the UMAP have smaller dimensions than at the top. For reference, the morphology of 4 astrocytes is also shown. **e** Boxplot summarizing the distribution of the angle of the major axis of astrocytes from different layers with respect to the pial surface ($n = 4763$ cells). Astrocytes from layer 2/3 are elongated perpendicularly with respect to the pial surface. For reference, the part of the UMAP representation corresponding to astrocytes colored by the cortical layer each also shown. Boxplots denote the median (line), interquartile range (box), $\min(\max(x), Q3 + 1.5 \, IQR)$ (upper whisker), and $\max(\min(x), Q1 - 1.5 \, IQR)$ (lower whisker). **f** The ability of CAJAL to identify morphological differences between neurons of different molecular type and T cells from different anatomical locations is evaluated in comparison to four general algorithms for cell morphometry and according to three different metrics of performance. The dashed line indicates the accuracy of a random classifier. CTS score: cell-type separation score. Source data are provided as a Source data file.

Moreover, using a fixed-step approach to sample points from the outline of the cells generally produced more accurate results than random sampling (Supplementary Fig. 2a). Expectedly, the accuracy of the predictions was stable against different choices of sampled points (<10% variation in MCC), and both accuracy and running time increased with the number of sampled points (Supplementary Fig. 2b–d). The accuracy of the predictions saturated at approximately 100 sampled points in these datasets, indicating no major advantage in sampling a larger number of points. However, we expect this number to depend on the morphological complexity of the cells and may differ in other datasets and cell types.

The results of these analyses collectively showcase the utility and versatility of the GW distance to perform unbiased studies of complex cell morphologies.

## GW cell morphology spaces recapitulate heterogeneous cell types

We next evaluated the ability of the GW distance to summarize cell morphologies across heterogenous cell populations. For that purpose, we used CAJAL to study the morphologies of 70,510 cells from a cubic millimeter volume of the mouse visual cortex profiled by the Machine Intelligence from Cortical Networks (MICrONS) program using two-photon microscopy, microtomography, and serial electron microscopy[43]. This dataset not only includes neurons, but also several types of glia and immune cells.

The UMAP representation of the cell morphology space produced by CAJAL recapitulated in an unsupervised manner the broad spectrum of cell types that are present in the tissue, including several populations of neurons, astrocytes, microglia, and immune cells (Fig. 3a). These populations were consistent with the manual annotations of 185 individual cells provided by the MICrONS program (Fig. 3b), except for a small number of neuronal reconstructions in the cluster of non-bipolar neurons that were manually annotated as bipolar. These manually annotated reconstructions exhibited a larger number of branches and higher cable length than manually annotated neurons in the cluster of bipolar neurons (Supplementary Fig. 3a, Wilcoxon rank-sum test $p$ value < 0.01). A closer examination indicated a possible origin of these discrepancies in the presence of segmentation artifacts in these reconstructions (Supplementary Fig. 3b).

Neurons from different cortical layers were associated with distinct regions of the cell morphology space, indicating the presence of morphological differences between neurons from different layers (Fig. 3c). In addition, our analysis uncovered morphological differences between astrocytes located in different cortical layers (Fig. 3d, e). Specifically, layer 1 astrocytes were substantially smaller, and layer 2/3 astrocytes were elongated perpendicularly to the pial surface, in comparison to astrocytes residing in other cortical layers (Fig. 3d, e). These differences were consistent with recent observations based on Glast-EMTB-GFP transgenic mice[44].

To quantitatively evaluate the ability of the GW distance to summarize cell morphologies across very different cell types in comparison to current general methods for cell morphometry, we considered the 3D morphological reconstructions of 512 T cells from the mouse popliteal lymph node, submandibular salivary gland, and skin, profiled with intra-vital two-photon microscopy[21], in addition to the Patch-seq dataset of the mouse motor cortex[38] described above. In contrast to neuronal morphometry methods, general cell morphometry methods can be applied to arbitrary cell types. Hence, we evaluated the ability of CAJAL and four other general methods (CellProfiler[45], SPHARM[21,24], Zernike moments[22,23], and the PCA-based approach of Celltool[22] and VAMPIRE[46]) to predict the anatomic location of each T cell and the transcriptomic class of each neuron based on their morphology. In this analysis, CAJAL performed similarly to the other methods in distinguishing the morphologies of T cells from different anatomic locations (Fig. 3f). However, the performance of the other methods in the analysis of neuronal morphologies was substantially inferior, and only CAJAL achieved a high accuracy and CTS score in both datasets (Fig. 3f). Overall, these results underscore the effectiveness of the GW distance in summarizing the vast range of cell shapes present in mammalian tissues.

## Multimodal analyses of GW cell morphology spaces enable uncovering genetic determinants of cell morphology

The combined analysis of morphological and genomic data from individual cells has the potential to unravel the genetic and molecular pathways that are associated with the progression of high-level cellular processes such as cell differentiation and plasticity. Since changes in cell morphology are continuous, establishing associations between cell morphology and other data is best accomplished by methods of analysis that are purpose-built for continuous processes. We extended our previous work on clustering-independent analyses of omics data[47] to implement a statistical approach for identifying molecular and physiological features that are associated with changes in cell morphology. We use the Laplacian score for feature selection[34] to test the association between the values of each feature and the structure of the morphology space, while accounting for user-specified covariates such as the age of the individual (Fig. 4a). To illustrate this approach, we used it to identify genes that contribute to the morphological plasticity of neurons in *C. elegans*. For that purpose, we considered the 3D morphological reconstructions of the DVB neuron in male worms. The DVB neuron is an excitatory GABAergic motor interneuron located in the dorso-rectal ganglion of the worm. It develops post-embryonically and undergoes post-developmental neurite outgrowth in males, altering its morphology and synaptic connectivity, and contributing to changes in the spicule protraction step of male mating behavior[48]. We applied our approach to identify loss of function mutations that are associated with changes in the dynamic morphology of the DVB neuron, taking the age of the worm as a covariate to reliably compare morphological changes across timepoints in adulthood. We considered 12 gene mutants, 5 double mutants, and controls across days 1 to 5 of adulthood (Supplementary Table 3), including 7 gene mutants and 1 double mutant from a previous study[49]. Our analysis identified mutations in the genes *unc-97, lat-2, nlg-1, unc-49, nrx-1*, and *unc-25* as

significantly affecting the morphology of the DVB neuron (Fig. 4b; Laplacian score permutation test, FDR < 0.05). Consistent with these results, cells from the same developmental stage or carrying one of the significant mutations appeared localized in the UMAP representation of the cell morphology space (Fig. 4c, d). The results of this analysis were stable against different choices for the scale parameter ($\varepsilon$) of the Laplacian score[47] and the addition of small amounts of noise to the digital reconstructions of the DVB neuron (Supplementary Figs. 4a, b). By repeating this analysis for worms of each age separately, we identified the age at which each of these mutations starts to significantly affect the morphology of the DVB neuron (Fig. 4e). To interpret these morphological differences in terms of neuronal characteristics, we used the Laplacian score to evaluate the association of 33 morphological features with the structure of the cell morphology space (Supplementary Dataset 1). We found that mutations in *nlg-1* and *unc-25* caused an increase in neurite length and number of branches compared to control worms (Supplementary Fig. 4c), while inactivating mutations in *unc-97* and *nrx-1* stunted neurite growth (Supplementary Fig. 4c). Altogether, these results are consistent with previous findings[49], where the morphological phenotype of inactivating mutations in *unc-97, nlg-1*, and *nrx-1* was described, and they extend them by uncovering new genetic determinants of neuronal plasticity in *C. elegans* and quantitative differences in the age of onset of the morphological alterations induced by different genes.

## Integrative analysis of molecular, physiological, and morphological data from single cells

The incorporation of single-cell RNA-seq onto whole-cell patch-clamp, known as Patch-seq, has enabled concurrent high-throughput measurements of the transcriptome, physiology, and morphology of individual cells[10]. The integrative analysis of these multi-modal data can be used to uncover transcriptomic and physiological programs associated with morphological differences between cells.

We used CAJAL to analyze the basal and apical dendrites of 370 inhibitory and 274 excitatory motor cortex neurons profiled with Patch-seq[38]. Consistent with our previous results, the GW cell morphology space captured morphological differences between the dendrites of neurons from different neuronal transcriptomic classes and cortical layers (Fig. 5a, b). By representing the pairwise distance between each pair of cells in the transcriptomic, electrophysiological, and morphological latent spaces as a point in a 2D simplex, we found a large degree of variability in the morphology of the dendrites of extratelecephalic-projecting (ET) neurons that was not paralleled by their gene expression profile (Fig. 5c). In contrast, the dendrites of *Lamp5*⁺ and bipolar (*Vip*⁺) GABAergic neurons showed limited morphological variability in comparison to their gene expression profile (Fig. 5c).

We characterized the gene expression and electrophysiological programs associated with morphological differences between neurons by using the Laplacian score approach described above. We performed this analysis separately for inhibitory and excitatory neurons to identify 173 and 556 genes, and 14 and 22 electrophysiological features, respectively, that were significantly associated with the morphological diversity of their dendrites (Fig. 5d and Supplementary Datasets 2 and 3; Laplacian score permutation test, FDR < 0.1). Expectedly, most of these genes (111/173 and 325/556) and electrophysiological features (14/14 and 22/22) were also associated with at least one transcriptomic type (gene-wise negative binomial generalized linear model and Wilcoxon rank-sum test, respectively, FDR < 0.1), indicating that most of these associations originate from morphological and electrophysiological differences between transcriptomic types and not necessarily from direct involvement in morphological processes. Nevertheless, among the 7 genes that were significant for both excitatory and inhibitory neurons, there were several genes that have been

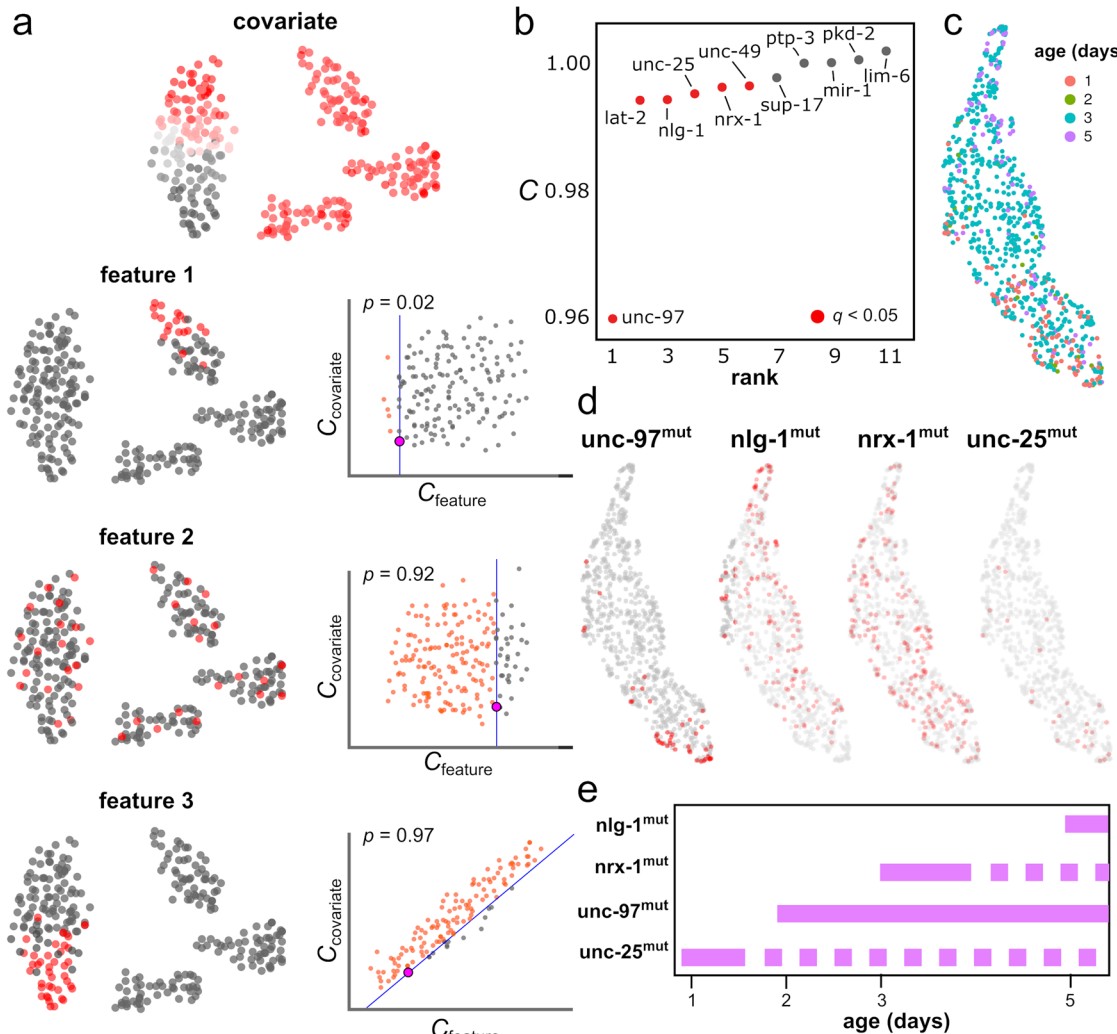

**Fig. 4 | Identification of mutations that have an impact in the morphology of an individual neuron. a** Schematic of approach for identifying features (gene expression, mutations, protein expression, etc.) associated with cell morphological changes based on multi-modal data. For each feature, the degree of consistency between the feature values and the structure of the cell morphology space is quantified using the Laplacian score ($C$). Features with a low score are associated with local regions of the cell morphology space. The statistical significance of each feature in relation to the covariates is evaluated by means of a one-sided permutation test. In the figure, examples of features that are significantly localized in the cell morphology space (feature 1, a small number of random configurations have a smaller value of $C_{feature}$, independently of the value of $C_{covariate}$), not significantly localized in the cell morphology space (feature 2, a large number of random configurations have a smaller value of $C_{feature}$), and substantially localized in the morphology space but in association with the covariate (feature 3, a small number of random configurations have smaller value of $C_{feature}$, but they are not independent on the value of $C_{covariate}$), are presented. **b** Mutations that have an impact on the morphology of the DVB interneuron in *C. elegans*. Null alleles are ranked according to their Laplacian score ($C$) in the cell morphology space of the DVB interneuron. The age of the worm was used as a covariate. Genes that significantly impact the morphology of the DVB interneuron are indicated in red (FDR < 0.05). **c, d** UMAP visualization of the cell morphology space of the DVB interneuron colored by the age of each worm (**c**) and the mutation status of *unc-97, nlg-1, nrx-1*, and *unc-25* (**d**) (red: mutated; gray: wild-type). **e** Restricting the analysis to worms of the same age allows us to identify the age of onset of the morphological effects induced by each significant mutation (FDR < 0.05). Dashed lines indicate time points for which there is limited data to restrict the analysis. Source data are provided as a Source data file.

previously reported to be involved in dendrite morphogenesis, such as *Dscam*, which plays a central role in dendritic self-avoidance[50], and *Pcdh7*, which regulates dendritic spine morphology and synaptic function[51].

We next investigated the presence of gene expression programs that form part of continuous morpho-transcriptomic trajectories. We computed the RNA velocity field in the gene expression space to predict the future gene expression state of each cell based on the observed ratio between un-spliced and spliced transcripts[52,53]. We reasoned that by projecting the RNA velocity field onto the GW cell morphology space and looking for transcriptomic trajectories that also appear as trajectories in this space, we could identify continuous transcriptomic processes that are correlated with changes in cell

morphology. Although RNA velocity trajectories were sparse in these data, consistent with the fact this dataset only consists of postnatal mice (ages 35 to 245 days)[38], our approach revealed several morpho-transcriptomic trajectories involving chandelier, basket, and *Lamp5*+ neurons (Fig. 5e), which might be the result of postnatal developmental programs and aging in mice. Cells along these trajectories showed increased complexity in their apical and basal dendrites in parallel to changes in their gene expression profile. To characterize these gene expression programs, we focused our analysis on 78 genes that were associated with the RNA velocity field of inhibitory neurons and computed the Laplacian score of each of these genes in the GW cell morphology space. This analysis revealed that 32 of the 78 RNA velocity genes were also significantly associated with the structure of

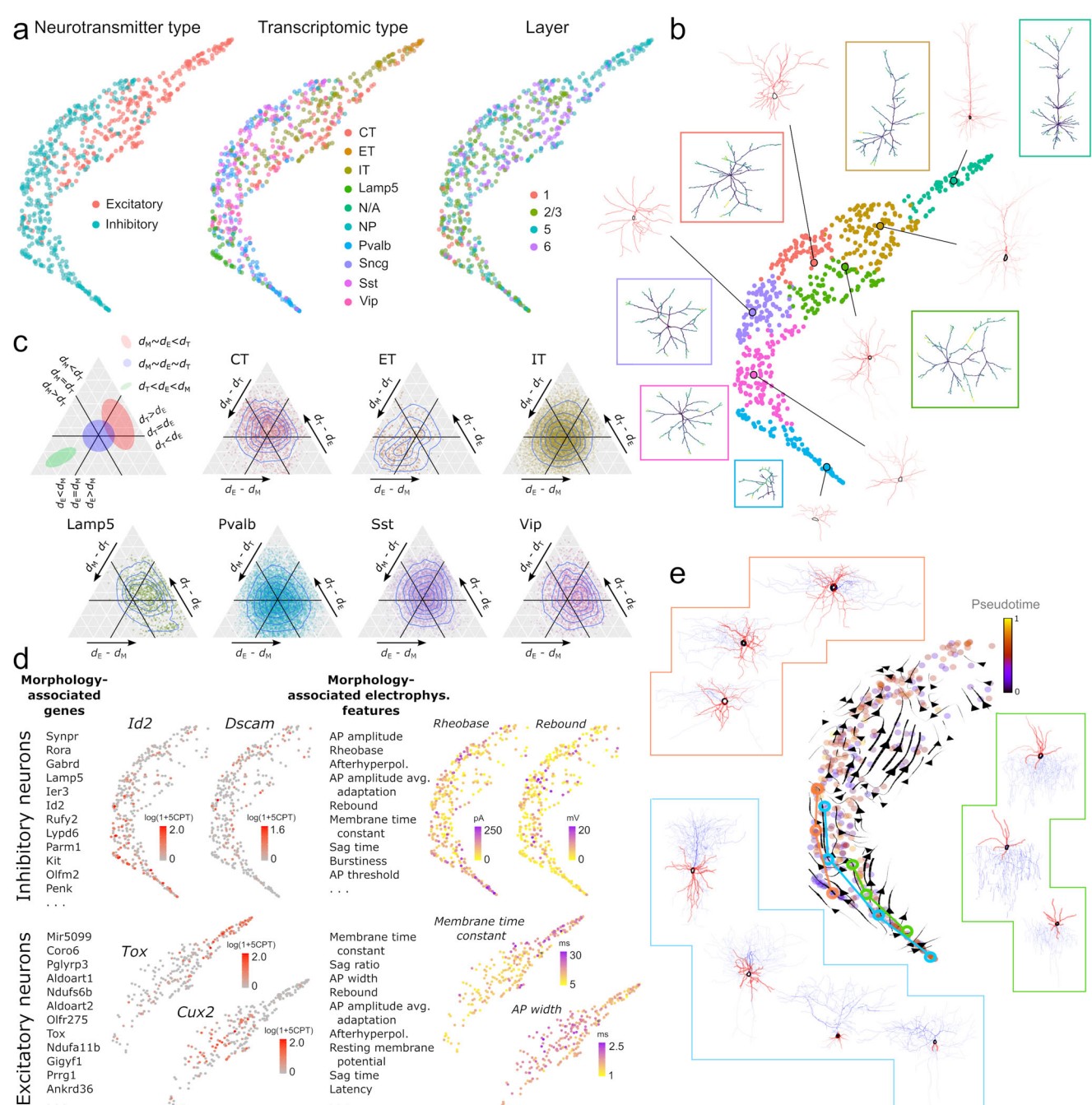

**Fig. 5 | Integrative analysis of molecular, physiological, and morphological data of mouse motor cortex neurons. a** UMAP representation of the GW cell morphology space of the dendrites of 370 inhibitory neurons and 274 excitatory neurons from the mouse motor cortex profiled with Patch-seq by Scala et al.[38]. The representation is colored by the neurotransmitter type (excitatory/inhibitory), the transcriptomic type, and the cortical layer of the cells, showing a large degree of localization of molecular and physiological features on the morphological space. **b** UMAP representation colored by the morphological cell populations defined by Louvain clustering. The medoid and average cell morphology (in boxes) are shown for each cell population. **c** Ternary plots showing the discrepancy between pairwise distances between cells in the morphology (M), transcriptomic (T), and electrophysiology (E) latent spaces for each transcriptomically defined population. The dendrites of ET excitatory neurons present a large degree of variability in their morphology which is not paralleled by consistent changes in their gene expression profile, whereas the dendrites of *Lamp5*⁺ GABAergic neurons present limited morphological variability in comparison to their gene expression profile. CT corticothalamic neurons, ET extratelecephalic neurons, IT intratelenchephalic neurons. **d** Top genes and electrophysiological features that are significantly associated with the morphological diversity of excitatory and inhibitory neurons according to their Laplacian score in the cell morphology space (FDR < 0.1). Most of the genes and electrophysiological features that are significantly associated with the morphological diversity of neurons are also associated with one or several t-types. The part of the UMAP corresponding to excitatory or inhibitory neurons is colored by the expression level and values of some of the significant genes and electrophysiological features, respectively. CPT: counts per thousand. **e** Morpho-transcriptomic trajectories computed by projecting the RNA velocity field in the cell morphology space. The morphology of several chandelier, basket, and *Lamp5*⁺ neurons along the trajectories is shown for reference.

the cell morphology space (Supplementary Dataset 4; Laplacian score permutation test, FDR < 0.05). These 32 genes included genes that code for secreted factors, such as *Spon1*, *Fgf13*, *Rspo2*, and *Reln* (Supplementary Fig. 5), and were enriched for genes involved in memory and cognition (GO enrichment adjusted *p*-value = 0.007). In contrast, the 46 genes that were associated with transcriptomic trajectories but not with morphological trajectories were enriched for genes involved in neuronal death and apoptosis (GO enrichment adjusted *p* value = 0.002).

Collectively, these results illustrate the utility of CAJAL to identify and characterize molecular and electrophysiological programs correlated with differences in cell morphology using single-cell Patch-seq data.

## GW cell morphology spaces facilitate the integration of cell morphology data across technologies

Advances in cell morphology profiling techniques have led to an explosion of high-resolution cell morphology data over the past decade[4]. The ability to perform integrated analyses of such data regardless of the experimental approach and technology that was used to generate them would be a powerful tool for imputing missing data and refining taxonomic classifications of cells. For example, by integrating patch-clamp and Patch-seq data the transcriptomic type of cells that have been profiled with patch-clamp could be imputed based on their morphology.

We used CAJAL to build a combined morphology space of the basal and apical dendrites of visual cortex neurons profiled with patch-clamp[35] and visual and motor cortex neurons profiled with Patch-seq[37,38]. The combined dataset consisted of 1662 neurons, of which 1,156 had associated single-cell RNA-seq data. Inhibitory and excitatory neurons from different datasets clustered together in separated regions of the combined morphology space (Fig. 6a), indicating that the structure of this space is mostly driven by biological differences rather than by experimental or computational artifacts. To evaluate the consistency of the combined cell morphology space, we considered the t-type[54] of the cells profiled with Patch-seq, and quantitatively assessed the distance in the combined cell morphology space between cells of the same transcriptomic class (for the Patch-seq data) or cells labeled with the corresponding Cre driver line (for the patch-clamp data). Cells of same transcriptomic class but from different Patch-seq datasets, as well as cells from the matching Cre driver line in the patch-clamp dataset, were closer to each other in the combined morphology space than cells from different transcriptomic classes or Cre driver lines (Fig. 6b, c and Supplementary Fig. 6; Wilcoxon rank-sum test *p* value < $10^{-100}$), demonstrating the utility of the GW distance for integrating cell morphology data across experiments and technologies.

We also used the same approach to refine the annotation of visual cortical neurons profiled with serial electron microscopy by the MICrONS program[43]. We considered 883 full neuron reconstructions from the two Patch-seq datasets and created a combined morphology space of these cells along with a subset of 1000 evenly sampled and 139 manually annotated neurons from the MICrONS dataset. As with the integration of patch-clamp and Patch-seq data, the manually annotated neuronal types from the MICrONS dataset were closer to Patch-seq cells of the matching t-type in the consolidated cell morphology space than to non-matching t-types (Fig. 6d, e and Supplementary Fig. 7; Wilcoxon rank-sum test *p* value < $10^{-100}$). For example, the only chandelier cell annotated in the MICrONS dataset was closer to *Pvalb Vipr2* t-type cells from the Patch-seq data than to cells from other t-types (Supplementary Fig. 7; Wilcoxon rank-sum test *p* value = 0.035).

Using this combined cell morphology space, we refined some of the manual annotations of the MICrONS dataset with more precise transcriptomic definitions. For example, of the three Martinotti cells

annotated in the MICrONS dataset, one cell presenting a distinct morphology with a densely arborized axon was closer in the morphology space to Patch-seq cells of the *Sst Chrna2* t-type (Fig. 6f; Wilcoxon rank-sum test *p* value = 0.045), while the other two Martinotti cells were closer to *Sst Calb2* t-type cells (Fig. 6f; Wilcoxon rank-sum *p* value = $10^{-3}$). This is consistent with previous results showing that expression of *Chrna2* is characteristic of layer 5 Martinotti cells that project into layer 1[55], and we confirmed that the soma of the predicted *Chrna2* Martinotti cell was indeed located in layer 5 while its long axon ended in layer 1 (Fig. 6f). Similarly, among the manually annotated basket cells in the MICrONS dataset, one had a more condensed morphology than the others (Fig. 6g). This smaller basket cell was close in the cell morphology space to *Vip Chat Htr1f* and *Vip Col15a1 Pde1a* t-type Patch-seq cells (Fig. 6g; Wilcoxon rank-sum *p*-value = 0.02), while larger basket cells were closer to *Pvalb Sema3e Kank4* and *Pvalb Gpr149 Islr* t-type Patch-seq cells (Fig. 6g; Wilcoxon rank-sum *p* value = $10^{-14}$). These results were again in agreement with the molecular characterization of small and large basket cells in the somatosensory cortex[56].

These results show the utility of GW cell morphology spaces to perform integrative analyses of cell morphological data across technologies and represent a conceptual basis for the development of algorithms for cell morphological data integration.

## Discussion

Shape registration has experienced several breakthroughs over the past 15 years with the formalization of new paradigms that allow for more flexibility in the quantification of morphology[57]. Here, we built upon one of these constructions, the GW distance, to develop a general computational framework and software for the multi-modal analysis and integration of single-cell morphological data. The proposed framework does not rely on predefined morphological features, is insensitive to rigid transformations, and can be efficiently used with arbitrarily complex and heterogeneous cell morphologies. Using this approach, we have accurately built, analyzed, and visualized cell morphology latent spaces. Like gene expression latent spaces in the analysis of single-cell RNA-seq data, morphological latent spaces are instrumental in the analysis and integration of single-cell morphological data. The metric properties of these spaces allowed us to integrate single-cell morphological data of neurons across experiments and technologies; identify morphological, molecular, and physiological features that define different subpopulations of neurons and glia; and establish associations between morphological, molecular, and electrophysiological cellular processes in these cells. Our quantitative and comparative studies using Patch-seq, patch-clamp, fMOST, electron, and two-photon microscopy data show that GW-based morphological analyses represent an improvement in accuracy and scope with respect to current methods for the analysis of cell morphology data, particularly in the Patch-clamp and fMOST datasets we analyzed, and enable previously unavailable analyses, such as inferring the transcriptional type of individual neurons based on the morphology of their dendrites or integrating morphological data across technologies.

The specific features of CAJAL in comparison to existing methods for cell morphometry can be informative in deciding which method to utilize for quantitative analysis of cell morphometry (Fig. 7). A defining characteristic of CAJAL is its versatility. Like methods based on simple geometric descriptors, GW-based cell morphometry can be used with arbitrary cell types (Fig. 3). Its power to discriminate complex morphologies, such as those of neurons and glia, is comparable, and in some cases superior, to that of cell-type specific descriptors like those produced by SNT[18] and L-Measure[17] (Fig. 2). In addition, similarly to moments-based descriptors, GW morphometry surveys cell shape from a physical perspective (by sampling points from the outline) instead of using user-defined lists of morphological features.

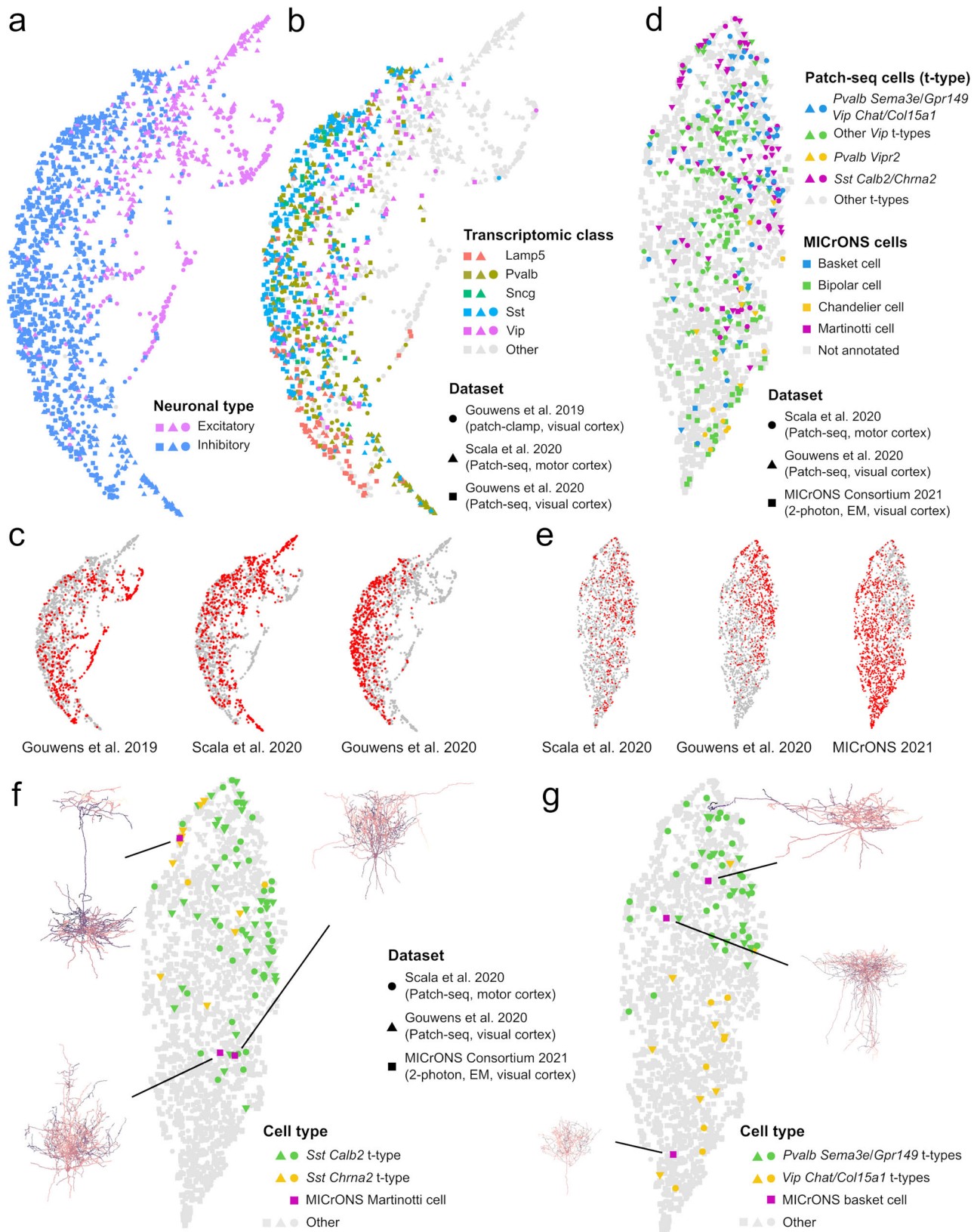

Although, the computational complexity of CAJAL is larger than that of methods based on user-defined lists of morphological features (in our studies, its application to 506 neurons using 100 sampled points per neuron took 80 min on a standard desktop computer), the implementation of recent strategies for reducing the computational complexity of GW distance[31,33] will substantially reduce the runtime of

CAJAL and thus enable sampling a larger number of points per cell in large datasets like the MICrONS dataset.

More generally, we expect that the analytic framework presented in this work will serve as the basis for the development of other currently missing computational methods for the analysis of single-cell morphological data, such as methods for batch-correcting cell

**Fig. 6 | Integration of cell morphology data across experiments and technologies. a** UMAP representation of the combined cell morphology space of the basal and apical dendrites of visual cortex neurons profiled with patch-clamp[35], and visual and motor cortex neurons profiled with Patch-seq[37,38]. The combined dataset consists of 1,662 neurons. The representation is colored by the neuronal type. **b** The same UMAP representation is colored by the Cre driver line (for patch-clamp cells) or the t-type (for Patch-seq cells). Cells of same t-type but from different Patch-seq dataset, and cells from the corresponding Cre driver line in the patch-clamp dataset, localize in the same regions of the morphology space. **c** The location of cells from each of the dataset is indicated in red in the UMAP representation of the combined cell morphology space. **d** UMAP representation of the combined cell morphology space of 883 full neuron reconstructions from the motor and visual cortices profiled with Patch-seq[37,38] and 1139 neurons from the mouse visual cortex with a combination of two-photon microscopy, microtomography, and serial

electron microscopy[43], 139 of which have been manually annotated by the MICrONS program. The manually annotated neuronal types from the MICrONS dataset localize in the same regions of the morphology space than Patch-seq cells from the corresponding t-type. **e** The location of cells from each of the datasets is indicated in red in the UMAP representation of the combined cell morphology space. **f** Refined annotation of 3 Martinotti cells that were manually annotated by the MICrONS program. One of the Martinotti cells presents a distinct morphology and is close in the morphology space to Patch-seq cells of the *Sst Chrna2* t-type, while the other two Martinotti cells are closer to *Sst Calb2* t-type cells. **g** Refined annotation of 3 basket cells that were manually annotated by the MICrONS program. One basket cell has a more condensed morphology than the others and is close in the morphology space to Path-seq cells of the *Vip Chat Htr1f* and *Vip Col15a1 Pde1a* t-types, while the other two larger basket cells are close to *Pvalb Sema3e Kank4* and *Pvalb Gpr149 Islr* cells.

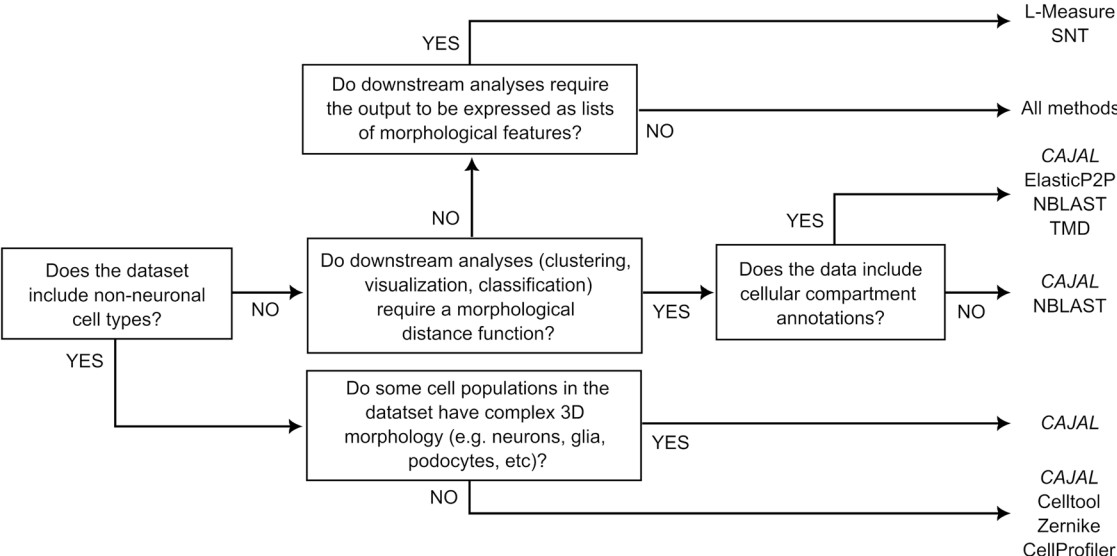

**Fig. 7 | Decision-tree-style summary of some of the main distinguishing characteristics between CAJAL and other methods for cell morphometry.** CAJAL and 8 other commonly used methods for cell morphometry are included.

morphological data or modeling morphological processes. We envision that the development of these and other methods for single-cell morphological analyses will significantly impact our understanding of the relation between morphological, molecular, and physiological diversification of cells.

## Methods

### Computation of GW cell morphology spaces

We build upon the application of metric geometry to the problem of finding a correspondence between two point-clouds such that the size of non-isometric local transformations is minimized[25–27]. CAJAL takes as input the digitally reconstructed cells. For each cell $i$, it samples $n$ points regularly from the outline and computes their pairwise distance matrix, $d_i$. It then computes the GW distance between every pair of distance matrices

$$\text{GW}(d_i, d_j) = \frac{1}{2} \min_{T_{ij} \in C} \sum_{\alpha,\beta,\delta,\gamma} |(d_i)_{\alpha\beta} - (d_j)_{\gamma\delta}|^2 (T_{ij})_{\alpha\gamma}(T_{ij})_{\beta\delta} \quad (1)$$

where the matrix $T_{ij}$ specifies a weighted pointwise matching between the points of cells $i$ and $j$, and $C$ represents the space of all possible weighted assignments[26]. By construction, CAJAL does not require pre-aligning cell outlines, since the input to $GW$ is the pairwise distance matrix within each cell, $d_i$, which is invariant under rigid transformations. Depending on the application, we

consider two choices for the distances $d_i$: Euclidean and geodesic distance.

The output is a metric space for cell morphologies which can then be clustered and visualized using standard procedures, such as Louvain community detection[36] and UMAP[58]. For each population of cells, $\mathcal{X}$, we compute its average morphology as the distance matrix

$$\left(\hat{d}_{\mathcal{X}}\right)_{\alpha\beta} = \frac{1}{\mathcal{X}} \sum_{i,\gamma} \left(T_{i\text{med}(\mathcal{X})}\right)_{\alpha\gamma} (d_i)_{\gamma\beta} \quad (2)$$

where med($\mathcal{X}$) denotes the medoid of $\mathcal{X}$ with respect to the $GW$ distance matrix. The morphology can then be visualized by computing the shortest-path tree or multidimensional scaling (MDS) of $\hat{d}_{\mathcal{X}}$. In addition, to facilitate the interpretation of morphology spaces, we find it useful to plot the values of standard morphological descriptors like cell height, width, diameter, neuronal depth, or fractal dimension[17,18,45] in the UMAP representation of the cell morphology space.

### Evaluation of features on the cell morphology space

To evaluate features, such as gene expression or electrophysiological properties, on GW cell morphology spaces, we build upon a spectral approach for clustering-independent analyses of multimodal data[34,47]. We first construct a radius neighbor graph of the $GW$ distance with radius $\varepsilon$. Each feature $g$ is represented by a vector $f_g$ of length the number of cells.

The Laplacian score of $g$ on the cell morphology space is then given by[34]

$$C_g = \frac{\sum_{ij} \left( \left(f_g\right)_i - \left(f_g\right)_j \right)^2 A_{ij}}{\mathrm{Var}\left(f_g\right)} \qquad (3)$$

where $A$ is the adjacency matrix of the radius neighbor graph, and $\mathrm{Var}(f_g)$ the estimated variance of $f_g$. Features with a low $C_g$ score are associated with morphologically similar cells. The significance of the score can be statistically assessed for each feature by means of a one-tailed permutation test and adjusted for multiple hypothesis testing using Benjamini–Hochberg procedure. To assess the significance of a feature $g$ in the presence of a set of covariates $h_m$, we perform a permutation test where the entries of $f_g$ and $f_{h_m}$ are simultaneously permuted and the scores $C_g$ and $\hat{C}_{h_m}$ are computed at each permutation. We denote these values collectively as $C_g^{\mathrm{null}}$ and $C_{h_m}^{\mathrm{null}}$. We then solve the regression problem

$$C_g^{\mathrm{null}} \sim \beta_0 + \sum_m \beta_m C_{h_m}^{\mathrm{null}} \qquad (4)$$

and consider the distribution of residuals as the null distribution for the adjusted score,

$$\widetilde{C}_g = C_g - \beta_0 - \sum_m \beta_m C_{h_m} \qquad (5)$$

## Processing of Patch-seq and patch-clamp morphological reconstructions

We downloaded the morphological reconstructions of neurons from several repositories. For the Patch-clamp dataset of Gouwens et al.[35], we downloaded 509 reconstructions in SWC format from the Allen Cell Types database, using Cell Feature Search and selecting for "Full" or "Dendrite Only" reconstruction types. Three of the SWC files were unsorted and were left out of further processing, for a total of 506 neurons. For the Patch-seq dataset of Gouwens et al.[37], we downloaded 574 reconstructions from the Brain Image Library (BIL) repository. We removed 62 neurons that did not have assigned transcriptomic types, for a total of 512 neurons. For the Patch-seq dataset of Scala et al.[38], we downloaded 645 reconstructions from the inhibitory and excitatory sets from the BIL repository, skipping the one inhibitory neuron that had no dendrites, for a total of 644 neurons. Lastly, for the fMOST dataset of Peng et al.[39], we downloaded 653 neuronal reconstructions from the BIL repository for which subclass or type annotations were available.

The SWC format represents each neuron as a tree of vertices, such that an edge can be drawn between a vertex and its parent, forming the skeleton of the neuron. From this format, we sampled 100 points radially around the soma at a given step size. We used a binary search to identify the step size which returns the required amount of evenly spaced points. To calculate the pairwise geodesic distance between these points, we constructed a weighted graph with weights given by the distance to the latest sampled point. We then used the Floyd–Warshall algorithm implemented in Networkx[59] to compute all pairwise shortest path distances in this graph. Alternatively, we computed the pairwise Euclidean distance between the 3D coordinates of these points.

We the computed the GW distance between each pair of cells as described above (subheading "Computation of GW cell morphology spaces") using the ot.gromov.gromov_wasserstein function of the "POT: Python Optimal Transport" Python library[60]. We then used this precomputed distance to build a 2D visualization of the morphology space using the https://github.com/tkonopka/umap package in R. We computed the Louvain clusters of a KNN graph of the GW distance using the multilevel.community function of the igraph R package[61].

## Average shape of neurons

To compute the average shape of a cluster of cells in the GW cell morphology space, we first found the medoid cell as the cell with the minimum sum of distances to all other cells in the cluster. To compute a morphological distance between cells, the GW algorithm identifies an optimal matching $T_{ij}$ between the points we have sampled (subheading "Computation of GW cell morphology spaces"). We used this matching to align the other cells in the cluster to the medoid, by reordering the pairwise geodesic distance matrix of their sampled points to match the distance matrix of the medoid cell. We rescaled the geodesic distance matrix of each cell into an unweighted graph distance by dividing out the minimum distance between any two points, so that the rescaled distances were integers. We set a threshold on these distances at 2, such that the distance was 0 from the point to itself, 1 to an adjacent point in the tree of the neuron trace, and 2 to any farther point. We averaged all of these distance matrices together over the cells in the cluster and built a $k = 3$ nearest neighbor graph, essentially connecting each sampled point to the three other points it was most often adjacent in the neurons of that cluster. We took the shortest path tree in this graph as the average shape for that cluster using Dijkstra's algorithm. We color each point in this average shape by a confidence value based on its minimum original unweighted graph distance, summed over the cluster, to any other point.

## Comparison of CAJAL with current methods for neuronal morphometry

We compared our approach to six other morphological methods for neuron analysis by applying them to the dendrite reconstructions of the neuronal datasets listed above (subheading "Processing of Patch-seq and patch-clamp morphological reconstructions"), except for the fMOST dataset, for which we considered the full reconstructions. These methods have stricter assumptions on the input, forcing us to remove disconnected components from the reconstructions. We applied NBLAST[19], as implemented in the nat.nblast R package (https://github.com/natverse/nat.nblast). We calculated a pairwise distance between all neurons using the nblast_allbyall function with the mean normalization method. We ran the Topological Morphology Descriptor (TMD) method of Kanari et al.[40] using the TMD Python package (https://github.com/BlueBrain/TMD). We followed their distances example (https://github.com/BlueBrain/TMD/blob/master/examples/distances_example.py) to compute the persistence image difference between every pair of neurons. We skipped 26 neurons across the two Patch-seq datasets for which get_ph_neuron or get_persistence_image_data errored due to a lack of bifurcating branches. We used the Measure Multiple Files batch script of the ImageJ SNT plugin[18] to compute morphological features of neurons, including the Sholl features[16]. We also computed morphological features using L-Measure[17], selecting all of their provided functions. We ran ElasticP2P[41] using their MATLAB implementation (https://github.com/50-Cent/ElasticPath2Path) with 500 samples per path.

We used three different metrics to assess the ability of these algorithms to identify morphological differences between Cre lines, transcriptomic types, or anatomic locations. We implemented the median-based group discrimination statistic used by Pincus and Theriot[22] to compare methods for cell-shape analysis. In addition, we used a 7-fold $k = 10$ nearest neighbor classifier from the scikit-learn Python library to predict the Cre driver line, transcriptomic class, or t-type of each cell based on morphological distance and used the Matthews correlation coefficient and the proportion of correctly classified cells to evaluate the accuracy of the predictions. Statistical significances were estimated based on 10 different random initializations of the 7-fold $k = 10$ nearest neighbor classifier.

## Morphological analysis of the MICrONS dataset

We downloaded the 113,182 static cell segmentation meshes from MICrONS using the trimesh_io module from the package MeshParty (https://meshparty.readthedocs.io/) at the lowest resolution (resolution 3). We then downloaded higher resolution meshes for cells that had less than 10,000 vertices at this lowest resolution. Cells with less than 1001 vertices at the lowest resolution were re-downloaded at the highest resolution (resolution 0). Cells with 1001 to 3000 vertices at the lowest resolution were re-downloaded at resolution 1, and cells with 3001 to 10,000 vertices were re-downloaded at resolution 2.

Along with other metadata available through the CAVEclient (https://github.com/seung-lab/CAVEclient), such as the 3D coordinates of neuron soma, we collected the cell IDs for each manually annotated cell type provided by the MICrONS program in their website. We used the layer 2/3, layer 4, and layer 5 manually annotated cells to estimate cortical layer boundaries in the y values of the 3D soma coordinates. We placed these cutoffs at layer 1 < 104,191 < layer 2/3 < 133,616 < layer 4 < 179,168 < layer 5 < 213,824 < layer 6.

We sampled 50 vertices from the triangular mesh of each cell, using the linspace function of the NumPy package[62] to evenly select vertices, since vertices were roughly ordered by proximity, and this gives an approximation of even sampling over the 3D space. We skipped the very large blood vessel mesh and 240 meshes with less than 50 vertices, for a total of 112,941 meshes. We used the heat method[63], implemented in the MeshHeatMethodDistanceSolver function of the Python library potpourri3D (https://github.com/nmwsharp/potpourri3d), to compute geodesic distances between the sampled points on the mesh. We parallelized the computation of the pairwise GW distance between the 112,941 meshes on 128 cores, but otherwise used the same process as with the Patch-seq and patch-clamp datasets (subheading "Processing of Patch-seq and patch-clamp morphological reconstructions"). Due to the large size of the resulting GW pairwise distance matrix, we used the Python libraries leidenalg[64] and umap-learn[65] to cluster the cells and compute 2D UMAP visualizations, respectively. We labeled the clusters based on the manually annotated cells provided by the MICrONS program.

We found that some morphological clusters mostly consisted of artifacts or neuron-glia doublets and removed those. In addition, another morphological cluster contained both neuron-neuron doublets and individual neurons with complex morphologies, so we devised an approach to remove meshes containing multiple somas from that cluster. We determined the number of somas in each mesh from that cluster by using MeshParty to skeletonize the meshes and convert them into graph representations where nodes have a radius value, and nodes within soma regions fall in a specific range of radii. For us, this range was 4000–30,000. We used HDBSCAN[66] to cluster these nodes in the 3D space and counted each cluster with at least three nodes as a soma. Meshes with more than one soma were removed from the cluster. Lastly, we noticed that many meshes with very high y coordinates appeared stretched, so we removed meshes with a y soma coordinate greater than 240,000. After removing all these artifacts, we recomputed a UMAP visualization of the remaining 70,510 cells in the cell morphology space using umap-learn.

For each astrocyte, we measured the bounding box by placing lower and upper bounds on the 1% and 99% quantiles of the mesh vertices along each of the first three principal components. We took the arccosine of the first principal component along the y axis to be the orientation angle of the astrocyte and measured its deviation from perpendicular.

## Morphological analysis of T cells

We retrieved 512 3D meshes of T cells from Medyukhina et al.[21]. We evenly sampled 200 points from the list of vertices in each mesh, which approximates an even sampling in 3D space since the vertices are roughly ordered in a spiral down the cell. We computed the GW distance of the pairwise Euclidean distances between these points as described above (subheadings "Computation of GW cell morphology spaces" and "Processing of Patch-seq and patch-clamp morphological reconstructions").

## Comparison of CAJAL with general methods for cell morphometry

We applied the Celltool method of Pincus and Theriot[22] using their Python package (https://github.com/zpincus/celltool). Since this method only works with 2D cell segmentations, we sampled the 2D boundary of the projection of each cell along the first two axes to the same number of sampled points used for CAJAL. We aligned these contours using a maximum of 20 iterations, allowing for reflections, and saved the non-normalized PCA values from the shape model. We used CellProfiler 4.0.3[14] on binary 2D projection images to compute both general shape features and Zernike moments using MeasureObjectSizeShape. We ran SPHARM[21] using their Python package (https://github.com/applied-systems-biology/Dynamic_SPHARM) on all of the mesh vertices for each cell. For neurons, we used the marching_cubes function of the scikit-learn Python library to define 3D mesh vertices. We used the spectrum.return_feature_vector function of SPHARM to extract the amplitude of harmonic components from the spectra produced by compute_spharm. We compared these methods to CAJAL using the same metrics described above (subheading "Comparison of CAJAL with current methods for neuronal morphometry").

## Evaluation of the accuracy and runtime of CAJAL as a function of the number of sampled points

We sampled 25, 50, 75, 100, and 200 points from each cell from the patch-clamp dataset of Gouwens et al.[35] and applied CAJAL as described above to compute the GW distance between cells. We used the Calinski–Harabasz score, the median-based statistic of Pincus and Theriot[22], and the Matthews coefficient of a $k = 10$ nearest neighbor classifier (subheading "Comparison of CAJAL with current methods for neuronal morphometry") to assess how the number of sampled points affects the ability of CAJAL to capture morphological differences between cells from different Cre driver lines. Runtimes were determined based on 12 threads of a desktop computer with an 8-core Intel Xeon E5-1660 3.20 GHz CPU.

## Morphological analysis of the DVB neuron

We considered the neurite reconstructions of the DVB neuron from 799 adult male *C. elegans* aged 1–5 days from control strains or strains containing mutations in the genes *nrx-1, mir-1, unc-49, nlg-1, unc-25, unc-97, lim-6, lat-2, ptp-3, sup-17*, or *pkd-2* (Supplementary Table 3). All strains were maintained on NGM-agar filled plates and seeded with OP50 *E. coli* bacteria as a food source at 23° Celsius. Strains used in this study were previously published[49] or were provided by the NIH Caenorhabditis Genetics Center (CGC). *C. elegans* were anaesthetized using 100 mM of sodium azide ($NaN_3$) and mounted on a pad of 5% agar on glass slides and analyzed by fluorescence microscopy on a Zeiss 880 confocal laser-scanning microscope. Fiji[67] was used to open Confocal **Z**-stack files and then loaded into the SNT[18] plugin. The DVB axon was traced from the soma to the ventral anterior turn, at the point it becomes a single anterior directed process. All neurites off this portion of the primary axon were included in the trace. SNT[18] was used to create and save 3D skeletons for input to CAJAL. We computed the GW distance between these morphological reconstructions as described above (subheadings "Computation of GW cell morphology spaces" and "Processing of Patch-seq and patch-clamp morphological reconstructions"), based on the geodesic pairwise distance of 100 points sampled from each neuron. We then introduced an indicator function for each mutated gene, which took values 0 or 1 on each cell depending on whether the worm had a wild-type or a mutated version

of the gene, respectively. To determine which of the 11 mutated genes were associated with changes in morphology, we computed the Laplacian score of each indicator function on the GW cell morphology space as described above (subheading "Evaluation of features on the cell morphology space"). To compute the score, we used the R package RayleighSelection[47] with 1000 permutations, $\varepsilon$ equal to the median GW distance and the age of the worm in days as a covariate. We evaluated the stability of the results against different choices for the scale parameter $\varepsilon$ by repeating this analysis for 11 different values of $\varepsilon$ corresponding to the 0th, 10th, 20th, ..., 80th, 90th, and 100th percentiles of the distribution of GW distances in the cell morphology space. We also evaluated the stability of the analysis against the addition of noise to the digital reconstructions of the DVB neuron. For that purpose, we shifted the coordinates of each vertex in the SWC files by random amounts sampled from a Gaussian distribution with mean 0 and standard deviation 1.5 and 7.2 mm, corresponding to 10% and 50% of the total standard deviation of the coordinates across the dataset, respectively. In the same way, we used RayleighSelection to determine which of 33 morphological features computed with SNT were significantly localized in the cell morphology space. In addition, we performed the same analysis using only neurons from a single day, for each day, to determine the age at which the effect of significant mutations on the morphology of the DVB neuron starts to emerge.

### Identification of genes and electrophysiological features associated with the morphology of neuronal dendrites

We used the same process described above (subheading "Processing of Patch-seq and patch-clamp morphological reconstructions") to compute the GW distance between the morphological reconstructions of the dendrites of 644 neurons profiled by Scala et al.[38]. We sampled 100 points from each dendrite and used geodesic distance to measure the distance between points. To determine which genes are associated with morphological variability we computed the Laplacian score of each gene on the GW morphology space using RayleighSelection, as described above (subheading "Evaluation of features on the cell morphology space"). Gene expression values were normalized as $\log(1 + 5000 \text{ size} - \text{normalized expression})$, we used 1000 permutations, and $\varepsilon$ was given by the median GW distance. We only tested genes expressed in at least 5% and less than 90% of cells. We identified gene ontology enrichments using the R package gProfileR[68], where we performed an ordered query of the significant genes based on their Laplacian score and restricted the search to biological process (BP) gene ontologies. We used the same procedure based on the Laplacian score to determine which electrophysiological features were associated with changes in the morphology of the dendrites.

### Computation of RNA velocity trajectories

We clipped 3' Illumina adapters and aligned FASTQ files to the GRCm38 mouse reference genome using the STAR aligner[69]. We used the command "run_smartseq" from the velocyto command line tool[52] to create a Loom file of spliced and unspliced reads. We then used the scvelo Python package[53] to compute RNA velocity trajectories. We tested scvelo in dynamical or stochastic mode with 0, 10, or 20 minimum counts; 500, 1000, or 2000 top variable genes; 10, 20, or 30 principal components; and 10 or 30 neighbors. We kept the velocity trajectories with the highest average confidence per arrow, defined by agreement with neighboring arrows. These trajectories were produced using stochastic mode with 0 minimum counts, 500 top variable genes, 10 principal components, and 30 neighbors. We computed the pseudotime using the velocity graph. We took all 78 genes which passed the basic default filters in rank_velocity_genes() to be velocity-related genes and used the Laplacian score to assess their morphological association.

### Consistency between transcriptomic, electrophysiological, and morphological spaces

We defined the transcriptomic distance ($d_T$) between two cells as the Spearman correlation distance between their log-normalized gene expression profile, and their electrophysiological distance ($d_E$) as the Euclidean distance between their electrophysiological feature vectors. We compared these distances and the GW morphological distance ($d_M$) between all pairs of cells in the dataset of Scala et al.[38] by representing them on a 2-simplex. For that purpose, we standardized the logarithm of pairwise distances independently for each data modality. We then took the axes of the 2-simplex to be the given by the difference between each pair of distances ($d_M - d_T$, $d_T - d_E$, $d_E - d_M$), so that the sum of the coordinates equals 0 for each pair of cells. We plotted cell pairs in the middle 98% of each axis.

### Integrative analysis of Patch-seq and patch-clamp data

We combined the patch-clamp and two Patch-seq datasets into one cell morphology space by computing the GW distance between the morphological reconstructions of the dendrites of all 1662 neurons from the 3 datasets. We sampled 100 points from each dendrite and used geodesic distance to measure distances between points.

To evaluate the integration of the Patch-seq datasets, we utilized the classification of neurons into the t-types of Tasic et al.[54]. This classification is provided by Gouwens et al.[37] as their transcriptomic alias, and we computed the classification for the dataset of Scala et al.[38] using their t-type-assignment Jupyter notebook. We tested the overlap between neurons of the same transcriptomic class but from different datasets in the cell morphology space by performing a Wilcoxon rank-sum test, comparing the distribution of GW distances within the same transcriptomic class with the distribution of GW distances between transcriptomic classes.

To evaluate the integration between the two Patch-seq datasets and the patch-clamp dataset, we matched the neuronal transcriptomic classes in the Patch-seq datasets with the Cre driver lines in the patch-clamp dataset. To define transcriptomic classes, we used the first marker in the t-types and considered markers that existed in at least five cells of two of the three datasets. This left Sst, Pvalb, and Vip as major markers between the transcriptomic classes and Cre lines, and Lamp5 and Sncg as markers between transcriptomic classes only. We again used the Wilcoxon rank-sum test to compare the distributions of GW distances within and between these five major transcriptomic types.

### Integrative analysis of Patch-seq and MICrONS neuronal data

We calculated a combined GW morphological space for the two Patch-seq datasets and 1000 neurons evenly sampled from the MICrONS dataset, in addition to 140 manually annotated neurons by the MICrONS program. We sampled 50 points from the full neuronal reconstructions from the Patch-seq datasets. In the case of the dataset of Scala et al.[38], this restricted our analysis to 370 neurons with full reconstructions. Since the SWC format used in the Patch-seq datasets contains a trace reconstruction, and the triangular cell segmentation meshes used in the MICrONS dataset contain cell surface reconstructions, we computed the GW distance based on the pairwise Euclidean distances between 50 points sampled from each neuron, instead of geodesic distance.

To evaluate the integration, we matched some of the manually annotated cells from the MICrONS dataset with t-types from the Patch-seq datasets. Following the results of Tasic et al.[54], we assigned the *Sst Calb2/Chrna2* t-types (*Sst Calb2 Pdlim5, Sst Calb2 Necab1, Sst Chrna2 Ptgdr, Sst Chrna2 Glra3*) to Martinotti cells, and the *Pvalb Vipr2* t-type to chandelier cells. Some other *Pvalb* t-types were assigned to basket cells, such as *Pvalb Sema3e Kank4* and *Pvalb Gpr149 Islr*, whereas CCK

or small basket cells were associated with *Vip* t-types such as *Vip Chat Htr1f* and *Vip Col15a1*[56]. Since cells of the *Vip* subclass have bipolar morphologies[54], we assigned all other *Vip* subtypes to bipolar cells. We then evaluated the consistency of the cell morphology space with these assignments by using a Wilcoxon rank-sum test to compare the distribution of GW distances between matching types across datasets with the distribution GW distances between non-matching types across datasets.

## Statistics and reproducibility

We considered all the available samples from the Patch-seq, intra vital two-photon microscopy, and MICrONS datasets used in this study. No statistical method was used to predetermine sample size. Sample sizes were chosen based on sample availability, correspond to some of the largest single-cell morphological datasets that are currently available, and led to statistically significant and reproducible results in our analyses. We excluded 3 neurons from the Gouwens et al. patch-clamp dataset[35] since their SWC file was not sorted. We excluded 62 neurons from the Gouwens et al. Patch-seq dataset[37] with no assigned transcriptomic type. The dendrites of one inhibitory neuron from the Scala et al. Patch-seq dataset[38] were not present and this neuron was therefore not considered in the analyses. 42,672 meshes from the MICrONS dataset were identified as artifacts or doublets after the filtering and quality control process detailed above and were therefore not considered in downstream analyses. For the DVB neuron morphology data, sample size was uniformly determined by analyzing at least 3 animals for each condition, and more than 10 animals in most cases. The order and allocation of controls and mutants was randomized for each replicate. The Investigators were not blinded to allocation during experiments and outcome assessment. Blinding to genotype allocation was not relevant to this study since all statistical tests involve the comparison between genotypes. We used two-sided Wilcoxon rank-sum tests to compare distances across conditions and one-sided permutation tests to evaluate the statistical significance of the Laplacian score. When testing multiple hypotheses, we used Benjamini–Hochberg procedure to control the false discovery rate.

## Reporting summary

Further information on research design is available in the Nature Portfolio Reporting Summary linked to this article.

## Data availability

All the datasets used in this study are publicly available. The morphological reconstructions of the DVB neuron generated in this study have been deposited in the neuromorpho.org database (Hart archive). The patch-clamp data of Gouwens et al.[35] are available at the "Allen Brain Atlas data portal [http://celltypes.brain-map.org/data]." The Patch-seq datasets of Gouwens et al.[37] and Scala et al.[38] are available at the Brain Image Library (BIL) using the URLs https://download.brainimagelibrary.org/biccn/zeng/pseq/morph/200526/ and https://download.brainimagelibrary.org/biccn/zeng/tolias/pseq/morph/, respectively. The fMOST dataset of Peng et al.[39] are available at the BIL using the URL https://download.brainimagelibrary.org/biccn/zeng/luo/fMOST/cells/. The two-photon microscopy data of Medyukhina et al.[21] are available at https://asbdata.hki-jena.de/publidata/MedyukhinaEtAL_SPHARM/. The MICrONS program dataset is available at the "MICrONS Explorer [https://www.microns-explorer.org/cortical-mm3#segmentation-meshes]." The GRCm38 mouse reference genome is available at https://www.ncbi.nlm.nih.gov/datasets/genome/GCF_000001635.20/. All other data supporting the findings of this study are available within the article and its Supplementary Files. Any additional requests for information can be directed to, and will be fulfilled by, the lead contact. Source data are provided with this paper.

## Code availability

The source code and documentation of CAJAL are available at https://github.com/CamaraLab/CAJAL and https://cajal.readthedocs.io/[28].

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

## Acknowledgements

The authors are grateful to Dr. Zhaolan Zhou for his constructive comments on the manuscript and Matthew Jozwik for assisting with the

conversion of files for the DVB analysis. This work was supported by the National Institute of Mental Health (NIMH) of the National Institutes of Health (NIH) under award number RF1 MH130553 (P.G.C.).

## Author contributions

K.G. performed the computational analyses. K.G. and P.N. implemented CAJAL. Y.L., J.C., and H.C. assisted with some of the computational analyses. A.B.S. implemented the covariate analysis of the Laplacian score. K.Z. and M.P.H. generated the morphological reconstructions of the DVB neuron and assisted with their analysis. P.G.C. and K.G. jointly wrote the manuscript. P.G.C. supervised the work.

## Competing interests

The authors declare no competing interests.
