## [Peer Review File · Nature Communications]

REVIEWER COMMENTS

Reviewer #1 (Remarks to the Author):

The authors of the manuscript entitled "Multi-modal analysis and integration of single-cell morphological data" describe a computational framework which extracts morphological information of pre-reconstructed neurons and glia with high accuracy and compares cellular morphology across different datasets. The authors use the anatomical classification to integrate morphological, transcriptomic and electrophysiological data to extract biophysical and molecular neuronal features that relate to the morphology of the neurons analyzed.

The manuscript is clearly written and includes references as appropriate. The necessity of morphological analysis pipelines is evident. The idea of capturing a higher order of complexity of neuronal anatomy by sampling the entire neuronal reconstruction of processes instead of focusing the feature extraction to arbitrarily chosen parameters (e.g. average branch length) gives high potential for anatomical classification. Moreover, the authors claim the CAJAL significantly outperforms already existing anatomical analysis packages. This statement is moderately supported by the provided data. Besides the novel strategy of anatomical analysis, incorporation of anatomical data into gene expression or electrophysiological analysis may bring up conceptual concerns. Specific concerns and suggestions regarding the scope and the message validity to enhance the impact of the work are below.

Major comments:

1: The authors used three different metrics (Calinski - Harabasz clustering score, the median-based group discrimination statistics and the Matthew's correlation coefficient) to validate the performance comparison of CAJAL to five already existing analysis packages. Though these metrics rely on published methods and existing clustering evaluation, it is unclear why the authors chose these specific metrics which do not demonstrate the power of the analysis directly. It would be highly beneficial to have a more direct and simplified comparison of the analysis packages for example with comparing the proportion/number of misclassified neurons by running CAJAL versus the five other already existing packages on the same pre-annotated data. This could also be done by any other comprehensible metrics (e.g analysis speed), though the authors mainly highlighted the classification accuracy as the advantage over the existing pipelines.

2: In order to validate that CAJAL's classification is satisfactory and recapitulates the manual classification of 506 neurons from the Gouwens et al., 2019 paper, the authors present a UMAP and describe similarities with:

“The resulting space of cell morphologies recapitulated the neuronal morphological types of the visual cortex (Fig. 2A). Cells with a similar morphology appeared in proximity in the UMAP representation of this space.”

Without a quantitative measure the reader is left with visual estimation of “proximity” on the UMAP to validate the accuracy of classification. This needs direct validation.

Similarly, another approach to validate CAJAL’s classification based on

“ A similar evaluation of the ability of each algorithm to identify 47 t-types of inhibitory neurons and 26 t-types of excitatory neurons in the motor cortex”

would not be convincing as the number of clusters found by CAJAL and each of the five already existing algorithms is not providing an accurate measure of the classification performance, since the number and the composition of clusters may depart from each other even in case of identical cluster number and differ across the six methods.

Similarly in Fig 3B, the authors show a visual representation of MICrONS clustering overlaid with manually annotated cells, which agree till some extent, though - if the coloring is correct on the figure - many of the “Bipolar cells” colored olive green, appear on the bottom of the “Neurons” cluster, but initially “Bipolar cells” were on the separate and well isolated cluster colored turquoise.

Also in this point just like above the proof of principle could be to simply compare the number of misclassified neurons to a pre-annotated dataset, and have an accuracy measure and compare this measure across the 6 analysis packages.

3: A major concept of the manuscript is to integrate morphological data into single-cell omics. This type of analysis attempts to align anatomical geometric properties of neurons with gene expression levels and electrophysiological parameters in an adult nervous system. The functional connection between these multimodal datasets is only existing if the data was obtained to address a change (e.g across conditions) within an already identified cell type as the authors did in the analysis of genes related to morphological plasticity of neurons in the *C. elegans*. In other words, the correlation of different modality features across different cell types is due to the identity of the different cell types hence, is lacking causality. Neuronal morphology is one of the major features of neuronal identity (first identified by Ramon y Cajal) and to-date we are not aware of cases where dendritic morphological difference - which would drive a classification - would not go together with electrophysiological identity. Similarly, as the authors can easily validate this in the used Patch-Seq datasets, looking at different morphological neuronal types would also go together with the transcriptome characteristic of the neuronal identity of the cell type under investigation. This lack of causality is further supported by looking at Fig.5D, where the authors lineup “Morphology-associated genes”. The vast majority of the highlighted genes are not encoding cytoskeletal proteins which could relate to the phenotype of dendritic structure. The genes listed are connected to a diverse set of cellular functions from axonal proteins (Synaptoporin (Synpr) or protocadherin 7 (Pcdh7)), through nuclear proteins (Ror-alpha (Rora) a nuclear hormone receptor),

endocytosis regulators (RUN And FYVE Domain Containing 2 (Rufy2)) and synaptic and lysosomal proteins (Lysosomal Associated Membrane Protein Family Member 5 (Lamp5)) up to cell-adhesion molecules (Down Syndrome Cell Adhesion Molecule (Dscam)) or neurotransmitter receptor subunits such as GABAA receptors delta subunit (Gabrd) ...etc. The function of the “Morphology-associated genes” is rather mixed and eclectic, but this broadness can be explained by the correlation to the neuronal type instead of the geometric shape of the neuron and thus the functional causality between these genes and the morphology.

In the case of intrinsic electrophysiology, the matter is similar. The intrinsic excitability parameters are independent of the dendritic morphology; they are another feature of the neuronal identity. There are neurons with the similar action potential which look extremely different, one can look at L2 pyramid cells and L5 pyramid cells, where the action potentials and firing patterns are similar but there is hundreds of microns of difference in their dendritic length and even soma sizes differ significantly.

Minor:

In Fig 3.B, the nomenclature is deceiving as “Bipolar cells” are also neurons, but “Neurons” are only labeled as the cells in the biggest cluster on the left along with other categories as “Microglia” or “Astrocytes” and “Bipolar cells”, if this is an anatomical nomenclature the use of “Non-Bipolar neurons” and “Bipolar neurons” could be more straightforward.

Fig 6B-C. is difficult to read as the different shapes of markers are blending together, perhaps the authors could consider to plot the datasets separate onto separate UMAPs for better visibility.

Taken together, if the authors would provide stronger support to their claim that the CAJAL package is efficient and especially that it outperforms the existing packages, it could be an extremely useful tool for the community. The crossmodal analysis of the morphology and gene expression was well-introduced with an example in the *C. elegans* experiments where the authors used the CAJAL to detect geometric malformations and explored the corresponding gene expression changes within the same cell-types. In this approach this computational framework could be used and optionally even further probed with for example distorted neuronal reconstructions to test whether it would pick up also slight or only robust malformations. However, the concept of incorporating geometrical neuronal measures with overall molecular and intrinsic electrophysiological features is rather intriguing and should be abandoned as different aspects of neuronal identity strengthen the multilayered definition of identity and does not give access to function-specific gene discovery.

Reviewer #2 (Remarks to the Author):

This paper establishes a general computational framework for the multi-modal analysis and integration of single-cell morphological data. The Gromov-Wasserstein (GW) distance is adopted to construct cell morphology latent spaces. Using these spaces, the morphological data across technologies are integrated and associated single-cell RNA-seq data and leveraged to infer relationships between morphological and transcriptomic cellular processes. In terms of applications, this framework is applied to imaging and multi-modal data of neurons and glia to uncover genes related to neuronal plasticity.

Here are some major comments in terms of the proposed methodology:

1. The authors mentioned several distance metrics in addition to GW distance, it is important to investigate the sensitivity of the proposed framework with respect to the distance metric.
2. Some other powerful cell shape analysis tools, such as the square-root vector fields (SRVF) [1], are also expected to be included in the analysis for comparison.

[1] Ximu Deng, Rituparna Sarkar, Elisabeth Labruyere, Jean-Christophe Olivo-Marin, Anuj Srivastava. Dynamic Shape Modeling to Analyze Modes of Migration During Cell Motility (2021)
<https://arxiv.org/abs/2106.05617>

3. I am curious about the proposed method for shape registration. This landmark-based shape registration methods usually face the issue in choosing the initial landmark. I would like to see if the proposed method can address this issue.
4. When evaluating the features on the cell morphology spaces, a neighbor graph of the GW distance is constructed with a specific radius. It is necessary to investigate if the hypothesis testing results are sensitive to the construction of neighbor graphs.

Reviewer #3 (Remarks to the Author):

The paper entitled Multi-modal analysis and integration of single-cell morphological data by Govek et al. the authors propose a new approach (under the name of CAJAL) to distinguishes cell morphology and

integrate cell morphology and sequence. They apply their method to neural cells and in the process recover genes associated with cell morphology.

The approach is very original and the results compelling. Claims are supported by the results. The authors make an extensive validation of their approach before identifying genes that regulate morphology. The manuscript is well written. I recommend the publication of this manuscript. I do have just few comments. Please address them before publication.

General comments

[page 4] The authors make several claims about the properties of their method but I did not see a discussion of these specific properties in the paper. Please justify these statements

- the approach has the generality and stability of simple shape descriptors
- the discriminative power fo cell-type specific descriptors
- unbiasedness and hierarchical structure of moments-based descriptors

[page 6] Description of the -omics part of CAJAL, and how omics and cell morphologies are combined is not included

[page 8] 100 points seems arbitrary. I assume this will depend on the curvature of the cells and also de number and length of dendrites. Is this correct?

[page 11] please briefly explain "GABAergic DVB interneuron"

minor comments

[page 3] Introduction -- lane 8, what do the authors mean by "this technique"

[page 5] "compact metric spaces" is not clear what this means, use italics?

[page 11] Figures 4C and D are not properly discussed in the main text

[page 14] "technical", do the authors mean experimental and/or computational?

[page 37] "lowe"

Response to the Reviewers' Comments

We appreciate the time and effort by you and the reviewers in providing constructive feedback about our manuscript. In the revised manuscript, we have carefully addressed all the comments and suggestions raised by the three reviewers. We describe in detail all the changes in the following point-by-point response. To facilitate the review, we also have highlighted in red major edits to the text.

Comments from Reviewer #1:

The authors of the manuscript entitled "Multi-modal analysis and integration of single-cell morphological data" describe a computational framework which extracts morphological information of pre-reconstructed neurons and glia with high accuracy and compares cellular morphology across different datasets. The authors use the anatomical classification to integrate morphological, transcriptomic and electrophysiological data to extract biophysical and molecular neuronal features that relate to the morphology of the neurons analyzed.

The manuscript is clearly written and includes references as appropriate. The necessity of morphological analysis pipelines is evident. The idea of capturing a higher order of complexity of neuronal anatomy by sampling the entire neuronal reconstruction of processes instead of focusing the feature extraction to arbitrarily chosen parameters (e.g. average branch length) gives high potential for anatomical classification. Moreover, the authors claim the CAJAL significantly outperforms already existing anatomical analysis packages. This statement is moderately supported by the provided data. Besides the novel strategy of anatomical analysis, incorporation of anatomical data into gene expression or electrophysiological analysis may bring up conceptual concerns. Specific concerns and suggestions regarding the scope and the message validity to enhance the impact of the work are below.

We thank the reviewer for their constructive comments and suggestions and the overall positive assessment of our manuscript.

Major comments:

1: The authors used three different metrics (Calinski - Harabasz clustering score, the median-based group discrimination statistics and the Matthew's correlation coefficient) to validate the performance comparison of CAJAL to five already existing analysis packages. Though these metrics rely on published methods and existing clustering evaluation, it is unclear why the authors chose these specific metrics which do not demonstrate the power of the analysis directly. It would be highly beneficial to have a more direct and simplified comparison of the analysis packages for example with comparing the proportion/number of misclassified neurons by running CAJAL versus the five other already existing packages on the same pre-annotated data.

The three metrics we use for evaluating performance capture complementary aspects of the association between morphological distances and molecular/physiological cellular types. Like the proportion of correctly classified cells (accuracy), Matthew's correlation coefficient is calculated from the confusion matrix directly. However, as opposed to accuracy, Matthew's correlation coefficient takes into account true and false positives and negatives and is therefore generally regarded as a more balanced measure than accuracy, which can be misleading with imbalanced data (see for example Chico and Jurman, BMC Genomics 21 (2020), <https://doi.org/10.1186/s12864-019-6413-7>). For this reason, we prefer Matthew's correlation coefficient to accuracy. However, we agree with the reviewer that accuracy can be more directly interpreted in some situations. To address this point, in Supplementary Figure 2 of the revised manuscript we now show the proportion of correctly classified cells, as well as the proportion of correctly classified cells that would result from random classification. By comparing this figure with Figures 2d and 3f, we see that the performance of CAJAL according to the proportion of correctly classified cells closely matches the results of Matthews's correlation coefficient.

This could also be done by any other comprehensible metrics (e.g analysis speed), though the authors mainly highlighted the classification accuracy as the advantage over the existing pipelines.

We focused on the ability of CAJAL to discern molecular/physiological cellular types based on their morphology because it is a non-trivial proxy of the amount of biological information captured by the morphological distances computed with different methods. Nevertheless, we agree with the reviewer that other information such as runtime is also useful. In the original manuscript, we included data about the runtime of our analyses (Supplementary Figure 3), but we did not put it in the context of other methods for cell morphometry. Overall CAJAL requires more computing time than feature-based cell morphology metrics, such as SNT. However, in practice, CAJAL can be still applied to very large datasets such as the MICrONS dataset (~113,000 cells), as we have shown in our manuscript. We hope to further reduce the runtime of the current implementation of CAJAL using distance imputation methods soon, and our preliminary results in this direction indicate that we can achieve a 200% reduction on runtime using this approach. To address this point in the revised manuscript, we have added some sentences in the Discussion section.

2: In order to validate that CAJAL's classification is satisfactory and recapitulates the manual classification of 506 neurons from the Gouwens et al., 2019 paper, the authors present a UMAP and describe similarities with:

"The resulting space of cell morphologies recapitulated the neuronal morphological types of the visual cortex (Fig. 2A). Cells with a similar morphology appeared in proximity in the UMAP representation of this space."

Without a quantitative measure the reader is left with visual estimation of "proximity" on the UMAP to validate the accuracy of classification. This needs direct validation.

We agree with the reviewer that a quantitative measure is preferred to support this statement in the manuscript. In the revised manuscript, we have compared the distribution of morphological distances computed with CAJAL for pairs of cells that belong to the same m-type according to

the morphological classification of (Gouwens et al., Nat. Neurosci. 22 (2019), <https://doi.org/10.1038/s41593-019-0417-0>) and for pairs of cells that belong to different m-types. Our analysis, summarized in the new Supplementary Figure 1 of the revised manuscript, shows that neurons of the same m-type are substantially closer to each other in the GW cell morphology space than neurons of different m-type (median fold change in distance = 3.44, Wilcoxon rank-sum test p -value $< 10^{-100}$).

Similarly, another approach to validate CAJAL's classification based on "A similar evaluation of the ability of each algorithm to identify 47 t-types of inhibitory neurons and 26 t-types of excitatory neurons in the motor cortex" would not be convincing as the number of clusters found by CAJAL and each of the five already existing algorithms is not providing an accurate measure of the classification performance, since the number and the composition of clusters may depart from each other even in case of identical cluster number and differ across the six methods.

Please note that in this analysis we are not clustering the cell morphology space produced by each method, which we agree with the reviewer would not provide a rigorous measure of performance. In (Scala et al., Nature 598 (2021), <https://doi.org/10.1038/s41586-020-2907-3>), two taxonomic levels for the transcriptomic classification of cells in the motor cortex were introduced: the transcriptomic class, and the transcriptomic subclass or t-type. In the analysis presented in Supplementary Figure 2b of the revised manuscript that the reviewer refers to, we are measuring the ability of each method to predict the specific t-type of each individual neuron based on their position in the cell morphology space. This is the same analysis as the one presented in the second column of Figure 2d with the only difference that in Supplementary Figure 2b we are evaluating the ability to predict the t-type of each cell (73 t-types) instead of the ability to predict the transcriptomic class of each cell (9 major transcriptomic classes). We have carefully reworded this part in the revised manuscript to improve clarity.

Similarly in Fig 3B, the authors show a visual representation of MICrONS clustering overlaid with manually annotated cells, which agree till some extent, though - if the coloring is correct on the figure - many of the "Bipolar cells" colored olive green, appear on the bottom of the "Neurons" cluster, but initially "Bipolar cells" were on the separate and well isolated cluster colored turquoise.

We have further investigated the origin of this discrepancy. Manually annotated bipolar neurons located in the cluster of non-bipolar neurons show indeed morphological differences with respect to manually annotated bipolar neurons located in the cluster of bipolar neurons, which suggest these differences are real and not an artifact from CAJAL. For example, in Supplementary Fig. 4a of the revised manuscript, we show significant differences in cable length and number of branches between these two groups of manually annotated bipolar neurons. Closer examination of these manually annotated cells reveals that these morphological differences likely originate from segmentation artifacts in some of the morphological reconstructions provided by the MICrONS Consortium. In particular, many of the manually annotated bipolar neurons that were incorrectly classified by CAJAL as non-bipolar neurons contain pieces from other nearby cells, as we show for several examples in Supplementary Fig.

4b of the revised manuscript. We have added some text in the revised manuscript (subheading “*GW cell morphology spaces recapitulate heterogeneous cell types*” of the Results section) to clarify this point.

Also in this point just like above the proof of principle could be to simply compare the number of misclassified neurons to a pre-annotated dataset, and have an accuracy measure and compare this measure across the 6 analysis packages.

We agree with the reviewer that a more quantitative metric of performance would be preferred in this example, similar to what we did with the datasets of Figures 2d and 3f and Supplementary Figure 2 where we consider the proportion of correctly classified cells as well as other metrics. However, please note that all the 7 methods for neuronal morphometry of Figure 2d, except for CAJAL, require neuronal traces as input and cannot be applied to non-neuronal cell types. Thus, none of those methods, except for CAJAL, can be directly applied to the MICrONS dataset, which consists of 3D meshes instead of neuronal traces and contains a mixture of neuronal and non-neuronal cell types. This limitation highlights one of the gaps in current methods for cell morphometry that our approach is filling in. We could instead apply some of the general methods for cell morphometry from Figure 3f. However, as we have argued above, the MICrONS dataset, as unique and useful resource it is, contains segmentation artifacts which would affect the metrics of performance in such an evaluation. For these reasons, we prefer to base our quantitative comparison with general methods for cell morphometry on the patch-clamp and 2-photon microscopy datasets of Figure 3f, for which we have reported the proportion of correctly classified cells in Supplementary Figure 2c of the revised manuscript.

3: A major concept of the manuscript is to integrate morphological data into single-cell omics. This type of analysis attempts to align anatomical geometric properties of neurons with gene expression levels and electrophysiological parameters in an adult nervous system. The functional connection between these multimodal datasets is only existing if the data was obtained to address a change (e.g across conditions) within an already identified cell type as the authors did in the analysis of genes related to morphological plasticity of neurons in the C. elegans. In other words, the correlation of different modality features across different cell types is due to the identity of the different cell types hence, is lacking causality. Neuronal morphology is one of the major features of neuronal identity (first identified by Ramon y Cajal) and to-date we are not aware of cases where dendritic morphological difference - which would drive a classification - would not go together with electrophysiological identity. Similarly, as the authors can easily validate this in the used Patch-Seq datasets, looking at different morphological neuronal types would also go together with the transcriptome characteristic of the neuronal identity of the cell type under investigation. This lack of causality is further supported by looking at Fig.5D, where the authors lineup “Morphology-associated genes”. The vast majority of the highlighted genes are not encoding cytoskeletal proteins which could relate to the phenotype of dendritic structure. The genes listed are connected to a diverse set of cellular functions from axonal proteins (Synaptoporin (Synpr) or protocadherin 7 (Pcdh7)), through nuclear proteins (Ror-alpha (Rora) a nuclear hormone receptor), endocytosis regulators (RUN And FYVE Domain Containing 2 (Rufy2)) and synaptic and lysosomal proteins (Lysosomal Associated Membrane Protein Family Member 5 (Lamp5))

up to cell-adhesion molecules (Down Syndrome Cell Adhesion Molecule (Dscam)) or neurotransmitter receptor subunits such as GABAA receptors delta subunit (Gabrd) ...etc. The function of the “Morphology-associated genes” is rather mixed and eclectic, but this broadness can be explained by the correlation to the neuronal type instead of the geometric shape of the neuron and thus the functional causality between these genes and the morphology.

It is not our intention to make any claim of causality in this section, which we agree with the reviewer cannot be supported with this type of data. We also agree that most of the associations between gene expression and morphology that we find in our analysis are expected to originate from morphological differences between neuronal types. However, additional transcriptomic, electrophysiological, and morphological variability within neuronal subtypes is expected from exogenous factors (see for example Hrvatin et al. Nat. Neurosci. 21 (2018), <https://doi.org/10.1038/s41593-017-0029-5>; Chevee et al. Cell Reports 22 (2018), <https://doi.org/10.1016/j.celrep.2017.12.046>; and Tripathy et al. PLOS Comput. Biol. 13 (2017), <https://doi.org/10.1371/journal.pcbi.1005814>). Although we agree that the Patch-seq dataset of (Scala et al. 2021) (as well as the other currently available Patch-seq datasets) is limited for this type of inference because it involves only one condition, we observe transcriptomic, electrophysiological, and morphological variability within individual t-types in these data, as also pointed out by (Scala et al. 2021) (section “Variability in individual t-types” in that paper). There are multiple factors that might contribute to this variability, including the broad range of mouse ages involved in the dataset (35 to 245 days).

Thus, although we agree with the reviewer on the limitations of the analysis presented in that section, we also find that this analysis uniquely illustrates some of the approaches enabled by CAJAL that will become particularly useful when data from developmental systems become more available. In the revised manuscript we have substantially edited this section to clearly state the limitations of the analysis. In addition, we have quantitatively compared the list of morphology-associated genes with the list of genes that are differentially expressed between t-types, showing that most of these associations are likely due to the morphological diversity of t-types and not necessarily to the involvement of these genes in morphological cellular processes.

In the case of intrinsic electrophysiology, the matter is similar. The intrinsic excitability parameters are independent of the dendritic morphology; they are another feature of the neuronal identity. There are neurons with the similar action potential which look extremely different, one can look at L2 pyramid cells and L5 pyramid cells, where the action potentials and firing patterns are similar but there is hundreds of microns of difference in their dendritic length and even soma sizes differ significantly.

We agree with the reviewer on this point and in the revised manuscript we have edited this section to clearly indicate the limitations of this analysis as described above.

Minor:

In Fig 3.B, the nomenclature is deceiving as “Bipolar cells” are also neurons, but “Neurons” are

only labeled as the cells in the biggest cluster on the left along with other categories as “Microglia” or “Astrocytes” and “Bipolar cells”, if this is an anatomical nomenclature the use of “Non-Bipolar neurons” and “Bipolar neurons” could be more straightforward.

We have fixed this in the revised manuscript.

Fig 6B-C. is difficult to read as the different shapes of markers are blending together, perhaps the authors could consider to plot the datasets separate onto separate UMAPs for better visibility.

We thank the reviewer for this suggestion. In the revised manuscript, we have more clearly indicated the distribution of the different datasets in the UMAPs of Figure 6b and c.

Taken together, if the authors would provide stronger support to their claim that the CAJAL package is efficient and especially that it outperforms the existing packages, it could be an extremely useful tool for the community. The crossmodal analysis of the morphology and gene expression was well-introduced with an example in the C. elegans experiments where the authors used the CAJAL to detect geometric malformations and explored the corresponding gene expression changes within the same cell-types. In this approach this computational framework could be used and optionally even further probed with for example distorted neuronal reconstructions to test whether it would pick up also slight or only robust malformations. However, the concept of incorporating geometrical neuronal measures with overall molecular and intrinsic electrophysiological features is rather intriguing and should be abandoned as different aspects of neuronal identity strengthen the multilayered definition of identity and does not give access to function-specific gene discovery.

We thank the reviewer for summarizing their comments and for the additional suggestion of distorting the neuronal reconstructions in the *C. elegans* analysis. In the revised manuscript we investigated the effect of adding Gaussian noise to the coordinates of the points in the digital reconstructions of the DVB neuron. Our results are shown in Supplementary Figure 5b for different amounts of noise, where we shifted the coordinates of each vertex in the SWC files by random amounts sampled from a Gaussian distribution with mean 0 and standard deviation 0, 1.5 and 7.2 μm , corresponding to 0% (no noise), 10%, and 50% of the total standard deviation of the coordinates, respectively. As expected, the addition of noise leads to larger pairwise distances between the neurons in the morphology space. However, the inferred associations between genetic defects and morphological changes are relatively robust against the addition of small amounts of noise. This is expected since the geodesic distances used for measuring pairwise distances between sampled points within each cell are largely stable if the topology of the neuron is not altered.

Comments from Reviewer #2:

This paper establishes a general computational framework for the multi-modal analysis and integration of single-cell morphological data. The Gromov-Wasserstein (GW) distance is

adopted to construct cell morphology latent spaces. Using these spaces, the morphological data across technologies are integrated and associated single-cell RNA-seq data and leveraged to infer relationships between morphological and transcriptomic cellular processes. In terms of applications, this framework is applied to imaging and multi-modal data of neurons and glia to uncover genes related to neuronal plasticity.

We thank the reviewer for summarizing the key aspects of our paper and for their constructive comments and suggestions.

Here are some major comments in terms of the proposed methodology:

1. The authors mentioned several distance metrics in addition to GW distance, it is important to investigate the sensitivity of the proposed framework with respect to the distance metric.

As we describe in the Results section of the manuscript (subheading “A general framework for the quantitative analysis of cell morphology data”), different metrics for measuring distances between sampled points within each cell lead to slightly different properties of the GW distance. These properties may be advantageous in different specific applications. Using a Euclidean metric to measure distances between sampled points in each cell gives rise to GW distances that account for the relative positioning of cell appendages. On the other hand, using a geodesic distance metric gives rise to GW distances that are invariant under bending deformations of the cell. As suggested by the reviewer, in the revised manuscript, we have investigated the sensitivity of CAJAL to the choice of distance metric using the neuron morphology data of (Gouwens et al., Nat. Neurosci. 22 (2019), <https://doi.org/10.1038/s41593-019-0417-0>). The results of this investigation are presented in Supplementary Fig. 3a of the revised manuscript. As it can be observed in that figure, both Euclidean and geodesic metrics led to accurate predictions of the transcriptomic class of each neuron. However, in these data, we observe that using a Euclidean metric generally gives rise to more accurate predictions than using a geodesic metric, indicating that the relative position of neuronal appendages, and not only their topology, contains useful information about the transcriptomic type of the cell.

2. Some other powerful cell shape analysis tools, such as the square-root vector fields (SRVF) [1], are also expected to be included in the analysis for comparison.

[1] Ximu Deng, Rituparna Sarkar, Elisabeth Labruyere, Jean-Christophe Olivo-Marin, Anuj Srivastava. Dynamic Shape Modeling to Analyze Modes of Migration During Cell Motility (2021) <https://arxiv.org/abs/2106.05617>

We thank the reviewer for bringing the square-root velocity (SRV) representation to our attention. This interesting mathematical framework can be used to compare the shape of 1-dimensional curves embedded in a higher-dimensional Euclidean ambient space based on the elastic distance (Srivastava et al., IEEE Transactions on Pattern Analysis and Machine Intelligence 33 (2011), <https://doi.org/10.1109/TPAMI.2010.184>). Compared to the Gromov-Wasserstein approach used in CAJAL, the SRV produces a parametric representation of the shape space, which can for instance be used to compute the morphologies interpolating two

cells. However, contrary to the Gromov-Wasserstein formalism, the SRV representation can only be used with planar (non-intersecting) 1-dimensional curves. Thus, whereas it is a powerful approach for comparing neuronal tracing data or 1-dimensional outlines of 2D cell segmentation masks without pre-aligning the cell by means of a rigid transformation, this approach is limited when analyzing very different 3D cell shapes/meshes like those found in the MICrONS dataset.

In the revised manuscript, we have included ElasticP2P (Batabyal and Acton, 25th International Conference on Image Processing (2018), <https://doi.org/10.1109/ICIP.2018.8451446>) in our comparisons of CAJAL with current methods for neuronal morphometry (Figure 2d and Supplementary Figure 2). ElasticP2P decomposes the neuronal tracing data of neurons into sets of 1-dimensional paths and uses the SRV representation to compare the shape of paths from each pair of neurons. We used the MATLAB code of <https://github.com/50-Cent/ElasticPath2Path> and 500 samples per path. In these analyses CAJAL generally offered better results than ElasticP2P in the three datasets we considered, as summarized in Figure 2d and Supplementary Figure 2 of the revised manuscript.

3. I am curious about the proposed method for shape registration. This landmark-based shape registration methods usually face the issue in choosing the initial landmark. I would like to see if the proposed method can address this issue.

In the revised manuscript, we have studied the dependence of the results of CAJAL on the set of sampled points from each cell. We have considered two approaches to sampling points from the outline of the cells: fixed-step sampling starting from some randomly chosen point, and uniform random sampling. The results are presented in Supplementary Figure 3a-c of the revised manuscript. These results show that fixed-step sampling, as implemented by default in CAJAL, generally gives rise to more accurate predictions of the transcriptomic type of each cell than random uniform sampling. Our results also show that the accuracy of the predictions is largely stable with respect to different choices of the initial sampled point in fixed-step sampling. Specifically, the coefficient of variation of Matthews correlation coefficient across different initializations is <10% (and similarly for the other metrics of performance), which generally is substantially smaller than the observed differences between methods for cell morphometry in our comparative study (Figure 2d). We have described these results in the Results section of the revised manuscript (subheading “GW cell morphology spaces accurately summarize complex cell shapes”).

4. When evaluating the features on the cell morphology spaces, a neighbor graph of the GW distance is constructed with a specific radius. It is necessary to investigate if the hypothesis testing results are sensitive to the construction of neighbor graphs.

We have investigated the dependency of the results of the Laplacian score on the radius parameter in previous publications (Govek et al. PLOS Comp. Bio. 15 (2019), <https://doi.org/10.1371/journal.pcbi.1007509>; Rabadan et al. Nat. Comm. 11 (2020), <https://doi.org/10.1038/s41467-020-17659-7>). These analyses showed that the results are generally stable against different choices for the radius parameter, except at the boundary of the parameter space where the radius is set to the minimum or maximum distance between points

in the point cloud. To show this in the context of cell morphology spaces, in the revised manuscript we have repeated the analysis presented in the subheading “*Multimodal analyses of GW cell morphology spaces enable uncovering genetic determinants of cell morphology*” of the Results section for 11 choices of the radius corresponding to the 0th, 10th, 20th, ..., 80th, 90th, and 100th percentiles of the distribution of GW distances in the cell morphology space. The results are summarized in the new Supplementary Figure 5a of the revised manuscript and show that the set of genes that are found to significantly affect the morphology of the DVB neuron is largely consistent across different choices of the radius parameter.

Comments from Reviewer #3:

The paper entitled Multi-modal analysis and integration of single-cell morphological data by Govek et al. the authors propose a new approach (under the name of CAJAL) to distinguishes cell morphology and integrate cell morphology and sequence. They apply their method to neural cells and in the process recover genes associated with cell morphology.

The approach is very original and the results compelling. Claims are supported by the results. The authors make an extensive validation of their approach before identifying genes that regulate morphology. The manuscript is well written. I recommend the publication of this manuscript. I do have just few comments. Please address them before publication.

We thank the reviewer for their constructive comments and suggestions and the positive assessment of our manuscript.

General comments

[page 4] The authors make several claims about the properties of their method but I did not see a discussion of these specific properties in the paper. Please justify these statements

- the approach has the generality and stability of simple shape descriptors*
- the discriminative power fo cell-type specific descriptors*
- unbiasedness and hierarchical structure of moments-based descriptors*

We thank the reviewer for pointing this out. In the revised manuscript, we have added a paragraph in the Discussion section justifying these statements. In summary, the results of our analyses in Figure 3 show that GW-based morphometry can be applied to arbitrary cell types, similarly to methods that are based on simple shape descriptors (for example, some of the descriptors implemented in CellProfiler). Moreover, as shown in Figures 2d and 3f of the manuscript, the power of GW-based morphometry to discriminate complex morphologies, such as those of neurons and glia, is comparable or superior to that of cell-type specific descriptors like those produced by SNT and L-Measure. Finally, similarly to moments-based descriptors, CAJAL surveys cell shape from a physical perspective by sampling points from the outline of the cell instead of using user-defined lists of morphological features.

[page 6] Description of the -omics part of CAJAL, and how omics and cell morphologies are combined is not included

We present most of the description of the omics part of CAJAL in subheadings “*Multimodal analyses of GW cell morphology spaces enable uncovering genetic determinants of cell morphology*” and “*An integrative analysis of molecular, physiological, and morphological data from single cells identifies continuous morpho-transcriptomic trajectories*” of the Results section. However, we agree with the reviewer that introducing the key aspects in subheading “*A general framework for the quantitative analysis of cell morphology data*” of the Results section would improve the presentation. In the revised manuscript, we have added some text at the end of that subsection where we briefly mention the use of the Laplacian score to establish associations between omics and morphological data.

[page 8] 100 points seems arbitrary. I assume this will depend on the curvature of the cells and also de number and length of dendrites. Is this correct?

The optimal number of sampled points will indeed depend on the complexity of the cell morphology. In our studies we found that sampling more than 100 points lead to limited increases in the accuracy of predicting the molecular type of neurons using patch-clamp and patch-seq morphological data (Supplementary Fig. 3b of the revised manuscript), whereas runtime increased substantially (Supplementary Fig. 3d of the revised manuscript). However, we expect this to depend on the specific dataset, as the reviewer mentions. For example, in our studies we found that for T cells a smaller number of points (~50 points) was sufficient to produce accurate predictions of their anatomical location. Similarly, we expect that for projection neurons, sampling a larger number of points may be beneficial. In the revised manuscript, we have added some text at the end of subheading “*GW cell morphology spaces accurately summarize complex cell shapes*” of the Results section to clarify this point.

[page 11] please briefly explain "GABAergic DVB interneuron"

In the revised manuscript, we have added some text in that subsection (subheading “*Multimodal analyses of GW cell morphology spaces enable uncovering genetic determinants of cell morphology*” of the Results section) to briefly explain the GABAergic DVB interneuron.

minor comments

[page 3] Introduction -- lane 8, what do the authors mean by "this technique"

We have replaced “this technique” with “patch-clamp” in the revised manuscript to improve clarity.

[page 5] "compact metric spaces" is not clear what this means, use italics?

Compact metric spaces are metric spaces that are complete (every Cauchy sequence of points in the space has a limit that is also in the space) and totally bounded (the space can be covered by finitely many subsets of fixed size). Mathematically the shape of any physical object is a compact metric space. Since the sentence where “compact metric spaces” appears refers to the mathematical definition of Gromov-Wasserstein distance, in the revised manuscript we have used italics for that term, as suggested by the reviewer.

[page 11] Figures 4C and D are not properly discussed in the main text

We thank the reviewer for pointing this out. We have added some text in this subsection (subheading “*Multimodal analyses of GW cell morphology spaces enable uncovering genetic determinants of cell morphology*” of the Results section) to properly introduce Figures 4c and d.

[page 14] "technical", do the authors mean experimental and/or computational?

To improve clarity, we have replaced “technical differences” by “experimental or computational artifacts” in the revised manuscript.

[page 37] "lowe"

We have fixed this typo in the revised manuscript.

REVIEWER COMMENTS

Reviewer #1 (Remarks to the Author):

I would like to thank the authors for their efforts to answer my questions, and to provide new data and figures presenting newly analyzed aspects of the performance of CAJAL. However, I have major concern about the efficacy and accuracy of the package based on the provided data. I start with citing my own comment from the summary of my initial review:

“Taken together, if the authors would provide stronger support to their claim that the CAJAL package is efficient and especially that it outperforms the existing packages, it could be an extremely useful tool for the community”

The revised data provided by the authors highlighted that the outstanding performance of the CAJAL package, especially that it outperforms other existing packages, was a strong overstatement.

I suggested to present a simpler validation of the classification performance of the CAJAL than indicated by the three different metrics (Calinski - Harabasz clustering score, the median-based group discrimination statistics and the Matthew's correlation coefficient) used by the authors. The actual meaning of these coefficients was not summarizing the performance in an easily understandable way, which could convince many readers, who could potentially use the CAJAL package. Therefore, I suggested showing the accuracy of classification using a pre-annotated dataset, by simply giving a percentage of correctly classified neurons and astrocytes from the used pre-annotated datasets. This task is mainly what the CAJAL could be used for.

The analysis (New Supplementary Figure 2a,b,c) showed less convincing results by far, compared to the originally presented figures (Fig. 2d and Fig. 3f (red bars)). In the original version in Fig. 2d, the first 2 coefficients demonstrated a 4-6-8-times increase in performance and the third coefficient shows a 1.5-2-times higher performance compared to any other package. When looking at the accuracy plots in Supplementary Figure 2a, in the first plot, we have differences of ~7% better than SNT, and around average performance on the second and third plot compared to other packages (maybe with a difference of 3-4%). In Supplementary Figure 2b, we see maximal of 1.5-times better performance than average but not outperforming all the packages with inhibitory neurons – which are extremely different in morphology - and in case of excitatory neurons it is similarly close to average (as in panel a) and not

outperforming the same package, TMD. Actually, TMD had higher accuracy to classify cells in 4/5 cases (panel (a) 2nd and 3rd plot and panel (b) both inhibitory and excitatory neurons).

This data overall shows that this package indeed works, but performs similarly to other packages, it may slightly outperform certain packages when used with a specific datasets, but it does not seem to be outstanding, as stated by the authors. The current manuscript, instead of providing convincing evidence that CAJAL is better at classification than other packages, is showing at best, that it is still better to run multiple packages, and look for a consensus across them, or simply use TMD, but no evidence is convincing enough to use CAJAL instead of TMD.

At this point, the major statement of the manuscript, that the CAJAL is outstanding and outperforming existing packages, became weak and for the rest of the manuscript, which is demonstrating different ways of comparing classifications from different datasets does not have a message validity which would convince the readership to use the CAJAL.

My other major comment is on the Multi-modal analysis:

The authors stated:

“we also find that this analysis uniquely illustrates some of the approaches enabled by CAJAL that will become particularly useful when data from developmental systems become more available.”

The multi-modal nature of the paper relies on the high efficacy of CAJAL, but as other packages could perform similarly, it is not clear what is specifically “enabled by CAJAL”; the multi-modal analysis is a goal and ongoing effort of many groups and as the authors stated here, when data compatible with such analysis becomes available, it is going to be useful to have high-accuracy classifiers, right now it is excessively challenging.

Reviewer #3 (Remarks to the Author):

The authors have addressed all my concerns and I recommend the publication of the manuscript.

Response to the Reviewers' Comments

We would like to express our gratitude to you and the three reviewers for the time and effort in providing us with constructive feedback on our manuscript. We are pleased to hear that reviewers #2 and #3 have recommended its publication, and we appreciate the additional valuable feedback from reviewer #1. In this revised version, we have carefully considered the new comments from reviewer #1 and have made the necessary changes in the manuscript to address them. To facilitate the review process, we have highlighted major edits in red throughout the manuscript. Along with these changes, we have released a new version of our software, which includes enhanced functionality and comprehensive documentation, available at <https://cajal.readthedocs.io/en/latest/index.html> .

Comments from Reviewer #1:

I would like to thank the authors for their efforts to answer my questions, and to provide new data and figures presenting newly analyzed aspects of the performance of CAJAL. However, I have major concern about the efficacy and accuracy of the package based on the provided data. I start with citing my own comment from the summary of my initial review:

“Taken together, if the authors would provide stronger support to their claim that the CAJAL package is efficient and especially that it outperforms the existing packages, it could be an extremely useful tool for the community”

The revised data provided by the authors highlighted that the outstanding performance of the CAJAL package, especially that it outperforms other existing packages, was a strong overstatement.

We appreciate the reviewer's feedback. We acknowledge that the revised benchmarking results may not have been presented with sufficient clarity in our previous version of the manuscript. To address this concern, we have made substantial edits to the manuscript and incorporated additional data, as we describe in detail below.

I suggested to present a simpler validation of the classification performance of the CAJAL than indicated by the three different metrics (Calinski - Harabasz clustering score, the median-based group discrimination statistics and the Matthew's correlation coefficient) used by the authors. The actual meaning of these coefficients was not summarizing the performance in an easily understandable way, which could convince many readers, who could potentially use the CAJAL package. Therefore, I suggested showing the accuracy of classification using a pre-annotated dataset, by simply giving a percentage of correctly classified neurons and astrocytes from the used pre-annotated datasets. This task is mainly what the CAJAL could be used for.

In response to the reviewer's feedback, we now provide clear explanations of each of the performance metrics used to evaluate CAJAL and other neuronal morphometry methods. We focus on three metrics: the cell type separation score of Pincus et al.², accuracy, and Matthews

correlation coefficient. We have removed the Calinski-Harabasz index from our evaluation since it is conceptually similar to the cell type separation score and thus redundant.

These three metrics are easily interpretable and provide complementary and equally important information on the performance of each method for neuronal morphometry. The cell type separation score measures the degree of separation between molecularly defined cell types in the cell morphology space and has been used in previous comparative studies of cell morphometry methods². Accuracy measures the percentage of correctly classified cells (in our case, by a k nearest neighbor classifier). Matthews correlation coefficient (also known as Phi coefficient) is a standard interpretable metric in classification performance evaluation which overcomes the known limitations of accuracy by considering both true and false positives and negatives. For example, the accuracy of a random classifier is generally >0% and can differ substantially between datasets (in our analysis, it ranges between 5% and 32%, depending on the dataset). However, the Matthews correlation coefficient of a random classifier is always zero. This allows for unbiased inferences about the performance of a method across multiple datasets. The Matthews correlation coefficient is well-known in the computational biology community and is often taught in introductory statistics courses. In the revised manuscript, we have referenced Chico and Jurman, BMC Genomics 21 (2020), <https://doi.org/10.1186/s12864-019-6413-7>, a well-known paper (>2,400 citations in the past two years) that describes the utility of Matthews correlation coefficient to a broader audience.

Because of their complementary nature, we believe that all three metrics (cell type separation score, accuracy, and Matthews correlation coefficient) are crucial for a comprehensive evaluation of CAJAL and other neuronal morphometry methods, and not just accuracy.

The analysis (New Supplementary Figure 2a,b,c) showed less convincing results by far, compared to the originally presented figures (Fig. 2d and Fig. 3f (red bars)). In the original version in Fig. 2d, the first 2 coefficients demonstrated a 4-6-8-times increase in performance and the third coefficient shows a 1.5-2-times higher performance compared to any other package. When looking at the accuracy plots in Supplementary Figure 2a, in the first plot, we have differences of ~7% better than SNT, and around average performance on the second and third plot compared to other packages (maybe with a difference of 3-4%). In Supplementary Figure 2b, we see maximal of 1.5-times better performance than average but not outperforming all the packages with inhibitory neurons – which are extremely different in morphology - and in case of excitatory neurons it is similarly close to average (as in panel a) and not outperforming the same package, TMD. Actually, TMD had higher accuracy to classify cells in 4/5 cases (panel (a) 2nd and 3rd plot and panel (b) both inhibitory and excitatory neurons).

In the revised manuscript, we have made improvements to facilitate the interpretation of the benchmarking analysis results. Instead of bar plots, we have presented the numerical quantification of the performance of each method in each dataset, in both absolute and relative terms, according to the three mentioned metrics. We have also statistically quantified the differences in performance between CAJAL and the other methods using a non-parametric test when a cross-validation scheme was possible. In addition, to provide a more comprehensive evaluation, we have expanded our analysis by including a recently published fMOST dataset by Peng et al.¹. Thus, the benchmarking analysis now includes four major datasets from three

different technologies (patch-clamp, Patch-seq, and fMOST). The results are presented in Figure 2d and Supplementary Tables 1 and 2 of the revised manuscript.

According to these results, *CAJAL* provides a 28% increase in accuracy with respect to SNT in the Patch clamp dataset (not 7%, as the reviewer states), and a 21% increase on average with respect to other methods in the two Patch-seq datasets (not 3-4%, as the reviewer states). We suspect that the reviewer may have interpreted absolute differences in accuracy instead of considering relative accuracies, which are more meaningful in this analysis. For instance, in the Patch-clamp dataset, the accuracy of *CAJAL* and SNT to predict the transcriptomic subtype of neurons based on their morphology was 29.4% and 22.9% respectively. Although the absolute difference between the two methods is only 6.5% (= 29.4% - 22.9%), this is not a meaningful metric since it is not possible to fully predict the transcriptomic subtype of a cell solely based on its morphology. In other words, the accuracy of a perfect cell morphometry method that achieves the maximum possible accuracy will be much lower than 100% and may not differ much in absolute value with respect to the accuracy of other methods. I.e., only relative accuracies are meaningful in this comparative analysis. For instance, in the above case, *CAJAL* represents a 28% increase in accuracy with respect to SNT ($29.4\% / 22.9\% = 1.28$), indicating a substantially better ability to capture biologically meaningful differences in cell morphology.

This data overall shows that this package indeed works, but performs similarly to other packages, it may slightly outperform certain packages when used with a specific datasets, but it does not seem to be outstanding, as stated by the authors. The current manuscript, instead of providing convincing evidence that CAJAL is better at classification than other packages, is showing at best, that it is still better to run multiple packages, and look for a consensus across them, or simply use TMD, but no evidence is convincing enough to use CAJAL instead of TMD.

As described above, in our revised manuscript we have considered four datasets and three different performance metrics (the cell-type separation (CTS) score of Pincus et al., accuracy, and Matthews correlation coefficient). We refer to Figure 2d and Supplementary Tables 1 and 2 for a detailed summary of the performance evaluation results.

Based on the CTS score, *CAJAL* overperformed all methods across all datasets, with a score that was on average 200% larger than that of other methods. Based on accuracy and Matthews correlation, *CAJAL* also demonstrated substantial improvement over all other methods, with a 42% and 84% increase in accuracy and Matthews correlation coefficient on average, respectively. While TMD showed comparable results to *CAJAL* in the two Patch-seq datasets (TMD provided a 0-4% increase in accuracy with respect to *CAJAL* in those two datasets), it performed substantially worse than *CAJAL* in the other two datasets (*CAJAL* provided a 56-61% increase in accuracy, and a 92-122% increase in Matthews correlation, with respect to TMD in those two datasets).

At this point, the major statement of the manuscript, that the CAJAL is outstanding and outperforming existing packages, became weak and for the rest of the manuscript, which is demonstrating different ways of comparing classifications from different datasets does not have a message validity which would convince the readership to use the CAJAL.

In our view, the quantitative results presented in Figure 2d and Supplementary Tables 1 and 2 demonstrate a solid improvement with respect to current methods for neuronal morphometry.

The additional functionalities provided by CAJAL, such as its applicability to arbitrary cell types and not just neurons, add to its usefulness. However, our objective is to maintain impartiality. Therefore, in the revised manuscript (section “*GW cell morphology spaces accurately summarize complex cell shapes*”), we have refrained from including any subjective interpretations or statements and solely focused on presenting the quantitative results of the benchmarking analysis.

My other major comment is on the Multi-modal analysis:

The authors stated:

“we also find that this analysis uniquely illustrates some of the approaches enabled by CAJAL that will become particularly useful when data from developmental systems become more available.”

The multi-modal nature of the paper relies on the high efficacy of CAJAL, but as other packages could perform similarly, it is not clear what is specifically “enabled by CAJAL”; the multi-modal analysis is a goal and ongoing effort of many groups and as the authors stated here, when data compatible with such analysis becomes available, it is going to be useful to have high-accuracy classifiers, right now it is excessively challenging.

We appreciate the reviewer's comment on the importance of the efficacy of CAJAL for the proposed multi-modal analyses. However, please note that CAJAL also introduces some novel methodology in this context, such as using the Laplacian score to identify features associated with morphology (Figure 4), quantifying discrepancies between morphological, electrophysiological, and transcriptomic characteristics of cells using 2-simplices (Figure 5), and predicting cell types by integrating morphological data across technologies for single-cell morphological profiling (Figure 6). While we could have implemented these novel analytical approaches on top of existing methods for cell morphometry, we believe that these strategies, in conjunction with the morphometric characteristics of the Gromov-Wasserstein distance, are of significant utility to the community.

Please also note that the statement quoted by the reviewer was only present in our rebuttal letter and not in the manuscript. Nevertheless, in the revised manuscript we have taken measures to moderate any assertions about the multi-modal analysis. In particular, we have eliminated any speculative statements from the last paragraph of the “*Integrative analysis of molecular, physiological, and morphological data from single cells*” section.

Overall, we believe that these revisions have greatly enhanced the quality of our manuscript by offering a more transparent depiction of CAJAL's performance in comparison to other methods. We hope the reviewer will find these revisions satisfactory and our updated manuscript to be informative and rigorous.

References

- 1 Peng, H., Bria, A., Zhou, Z., Iannello, G. & Long, F. Extensible visualization and analysis for multidimensional images using Vaa3D. *Nat Protoc* **9**, 193-208, doi:10.1038/nprot.2014.011 (2014).
- 2 Pincus, Z. & Theriot, J. A. Comparison of quantitative methods for cell-shape analysis. *J Microsc* **227**, 140-156, doi:10.1111/j.1365-2818.2007.01799.x (2007).

REVIEWER COMMENTS

Reviewer #1 (Remarks to the Author):

Dear Editor,

The authors of the manuscript entitled "Multi-modal analysis and integration of single-cell morphological data" describe a computational framework which extracts morphological information of pre-reconstructed neurons and glia with high accuracy and compares cellular morphology across different datasets. The authors use the anatomical classification to integrate morphological, transcriptomic and electrophysiological data to extract biophysical and molecular neuronal features that relate to the morphology of the neurons analyzed.

Overall, the interpretation of the same performance of the CAJAL package improved a lot, the performance of the package was not convincing to replace other packages and to be generally used, only on specific datasets, but it is not clear what makes the CAJAL performing better or worse on certain datasets and not on others. The original bar plot visualization of the accuracy performance demonstrated well the actual performance of the CAJAL, which made it clearer that CAJAL mostly performed similar to other packages on Patch-Seq datasets, performed better by ~5% in the Patch-Clamp dataset and in the fMOST it performed ~8% better than SNT. The audience who would consider using the package will not get better results by exaggeration of percentages normalized to each other, only compares the difference of the percentages.

To the authors:

In the previous revision I suggested using more simple metrics as the CAJAL package would not only be made for the members of the computational community. The authors used ground-truth pre-annotated datasets to test how well the extracted morphological information by multiple existing packages could predict the neuronal type. This metric was called "Accuracy" (proportion of correctly classified cells in %). The results of this metric across datasets and existing packages were intriguing rather than convincing. On the attached "png" red circles show where CAJAL did not outperform other packages in Accuracy and blue shows where CAJAL did outperform the existing packages, panel a by ~5% and in panel c by ~2-3%.

In the latest version of the revised manuscript:

The authors further improved the interpretation of the performance of the CAJAL package, and they came up with using another metric, the CTS score (Pincus et al. 2007) to prove the high accuracy of CAJAL to capture pre-reconstructed neuronal morphology. I am not convinced that the use of another

new metric improved the actual performance of the package. As the new metric is strongly in favor of the authors' narrative, the authors should explain what feature made this robust increase in the estimated performance compared to earlier metrics. The Accuracy metric is the most convincing and most tangible for those laboratories who are working with anatomy, Patch-Clamp electrophysiology or Patch-Seq.

The authors removed the bar plot figure of Accuracy and instead, tables were used to prove the performance of CAJAL. The authors implemented minor changes to the methods of calculation of the Accuracy metric of CAJAL (10 different random initializations of the 7-fold $k = 10$ nearest neighbor classifier) and included a new fMOST dataset. On the fMOST dataset, CAJAL performed better in average, compared to existing packages, thus it somewhat pulled down the average relative accuracy computed against the accuracy of CAJAL. This created the values in Figure 2d, which are very convincing that CAJAL does outperform other existing packages. However, most of the improvements were due to the inclusion of the fMOST dataset, and the averaging of relative percentages. The original figure presented as Supplementary Figure 2 (red bar plots used above on page 1), though, looked less convincing in terms of proving that CAJAL outperforms other packages, presented the comparison correctly, without exaggeration by normalized percentages. The authors neatly included all the original values of accuracies across each dataset and method in the new Supplementary Table, which is very useful, though a simple plot makes it much easier to comprehend and compare the results. The summary of the accuracy in main Figure 2d only became more convincing because of averaging and the inclusion of the fMOST dataset. I suggest showing the original accuracy plots - instead of table of numbers - for each dataset as the authors presented in the previous version of their manuscript, together with the fMOST dataset as a 4th bar plot next to the Patch-clamp and the two Patch-Seq comparisons (as in the bar plot above on page 1) without normalization to CAJAL as 100%. Separate plot for each used dataset would show the ground-truth of the performance of CAJAL.

Such an interpretation would show the same data, but it would also draw the attention of the readers that Accuracy strongly depends on the chosen dataset. As the authors found the fMOST dataset, where the CAJAL performed well, it indeed supported the usefulness of the CAJAL within that dataset. At the same time, we could observe in the originally used Patch-Seq datasets that CAJAL did not outperform existing packages. The authors should explain what the characteristic of a dataset is (brain region, cell types or a reconstruction method, other technical factor...etc) what makes the CAJAL outperform other packages, this would explain the advantage of the package and guide the potential future users to specifically apply it on a certain type of dataset.

High performance is one of the major statements of the manuscript. The authors explained that they normalized the values of accuracy to CAJAL's performance. The Accuracy by itself is a percentage, expressing that out of 100 neurons in case of their example, accuracy of SNT would be ~23 cells and accuracy of CAJAL would be ~29 cells. This simply means that CAJAL would score 29 instead of 23, which may be convincing to use the package, as it is truly an improvement. Normalization shows 1.28, which means 28% improvement, which is deceiving in this form. Without this normalization of percentages,

the performance of CAJAL does look better in certain cases but does not look outstanding on the accuracy plots I suggested, which are presenting ground-truth without exaggeration.

This being said, I appreciate the effort of the authors to improve the interpretation of the performance of CAJAL. The authors proved that the CAJAL works per se, and on certain datasets could outperform some existing packages by a given percentage. At the same time, it is clear that the CAJAL is only superior to some existing packages on certain datasets, and for some reason not when used on Patch-Seq anatomy.

Response to the Reviewers' Comments

We would like to thank you and the reviewer for the time and effort in providing us with constructive feedback. In this revised version, we have considered the additional comments from reviewer #1 and have made the necessary changes in the manuscript to address them, as we describe in the following point-by-point response.

Comments from Reviewer #1:

The authors of the manuscript entitled "Multi-modal analysis and integration of single-cell morphological data" describe a computational framework which extracts morphological information of pre-reconstructed neurons and glia with high accuracy and compares cellular morphology across different datasets. The authors use the anatomical classification to integrate morphological, transcriptomic and electrophysiological data to extract biophysical and molecular neuronal features that relate to the morphology of the neurons analyzed.

Overall, the interpretation of the same performance of the CAJAL package improved a lot, the performance of the package was not convincing to replace other packages and to be generally used, only on specific datasets, but it is not clear what makes the CAJAL performing better or worse on certain datasets and not on others. The original bar plot visualization of the accuracy performance demonstrated well the actual performance of the CAJAL, which made it clearer that CAJAL mostly performed similar to other packages on Patch-Seq datasets, performed better by ~5% in the Patch-Clamp dataset and in the fMOST it performed ~8% better than SNT. The audience who would consider using the package will not get better results by exaggeration of percentages normalized to each other, only compares the difference of the percentages.

Please note that our aim is not to exaggerate any differences, but to provide a fair and transparent assessment of our method's performance. As we detail in Supplementary Table 2 of the manuscript, the accuracies of SNT in the 4 datasets were 22.9%, 44.8%, 57.3%, and 57.9%, while the accuracies of CAJAL in these datasets were 29.4%, 51.5%, 63.9%, and 65.8%. The accuracy of CAJAL was therefore 1.12 – 1.28 times larger than that of SNT, depending on the dataset.

As detailed in our previous response letter and in the manuscript, relative accuracies are important for making inferences about the performance of each method across datasets. For example, the accuracy of a random classifier in predicting the molecular subtype of neurons based on their morphology is >0% and varies across datasets. Similarly, due to the inherent biological limitations in predicting molecular phenotypes from morphology, the accuracy of a perfect cell morphometry method would be <100% and vary across datasets.

In our manuscript we have presented the complete results of our benchmarking analysis, including both the absolute and relative quantification of accuracy (Figure 2d and Supplementary Table 2), so that the readers have comprehensive information about the performance of each method in each dataset. Following the suggestion by the reviewer, we do not show relative accuracies in the main figure and have replaced this panel with the original bar

plot visualization of accuracy (updated to include the results of the fMOST dataset). In addition, we have stated the absolute (non-normalized) accuracies of CAJAL and TMD in the main text. Finally, we have provided specific guidelines on when to use CAJAL, as we describe below.

In the previous revision I suggested using more simple metrics as the CAJAL package would not only be made for the members of the computational community. The authors used ground-truth pre-annotated datasets to test how well the extracted morphological information by multiple existing packages could predict the neuronal type. This metric was called "Accuracy" (proportion of correctly classified cells in %). The results of this metric across datasets and existing packages were intriguing rather than convincing. On the attached "png" red circles show where CAJAL did not outperform other packages in Accuracy and blue shows where CAJAL did outperform the existing packages, panel a by ~5% and in panel c by ~2-3%.

In the latest version of the revised manuscript:

The authors further improved the interpretation of the performance of the CAJAL package, and they came up with using another metric, the CTS score (Pincus et al. 2007) to prove the high accuracy of CAJAL to capture pre-reconstructed neuronal morphology. I am not convinced that the use of another new metric improved the actual performance of the package. As the new metric is strongly in favor of the authors' narrative, the authors should explain what feature made this robust increase in the estimated performance compared to earlier metrics. The Accuracy metric is the most convincing and most tangible for those laboratories who are working with anatomy, Patch-Clamp electrophysiology or Patch-Seq.

We would like to clarify that the CTS score, previously referred to as "Celltool score", was in the original version of the manuscript and has remained in all subsequent revisions. It is not a newly added metric, as the reviewer suggests. As we state in the manuscript (subsection "GW cell morphology spaces accurately summarize complex cell shapes"), we chose to include the CTS score because it quantifies the level of separation between molecularly defined cell types in the cell morphology space, and therefore provides complementary information to the other two metrics we use (accuracy and Matthews' correlation coefficient). In addition, this score was the main evaluation metric used in a previous comparative study of cell morphometry methods¹, making it an informative metric to include in our analysis.

The authors removed the bar plot figure of Accuracy and instead, tables were used to prove the performance of CAJAL. The authors implemented minor changes to the methods of calculation of the Accuracy metric of CAJAL (10 different random initializations of the 7-fold k = 10 nearest neighbor classifier) and included a new fMOST dataset. On the fMOST dataset, CAJAL performed better in average, compared to existing packages, thus it somewhat pulled down the average relative accuracy computed against the accuracy of CAJAL. This created the values in Figure 2d, which are very convincing that CAJAL does outperform other existing packages. However, most of the improvements were due to the inclusion of the fMOST dataset, and the averaging of relative percentages.

Note that the additional dataset (requested by the editorial office) only resulted in moderate changes to the average accuracy values. Specifically, we observed slight decreases in the accuracies of NBLAST, TMD, Sholl, and ElasticP2P normalized with respect to CAJAL, which changed from 67.6% to 63.6%, 89.3% to 82.5%, 72.7% to 65.5%, and 63.7% to 57.2%,

respectively. On the other hand, the average accuracies of L-Measure and SNT normalized with respect to CAJAL slightly increased from 67.6% to 68.3% and from 84.8% to 85.6%, respectively, after including the new dataset. These changes are within the expected range given the variability of different datasets. Hence, the inclusion of the additional dataset provides a more comprehensive evaluation of our method and confirms the overall conclusions of our study.

The original figure presented as Supplementary Figure 2 (red bar plots used above on page 1), though, looked less convincing in terms of proving that CAJAL outperforms other packages, presented the comparison correctly, without exaggeration by normalized percentages. The authors neatly included all the original values of accuracies across each dataset and method in the new Supplementary Table, which is very useful, though a simple plot makes it much easier to comprehend and compare the results. The summary of the accuracy in main Figure 2d only became more convincing because of averaging and the inclusion of the fMOST dataset. I suggest showing the original accuracy plots - instead of table of numbers - for each dataset as the authors presented in the previous version of their manuscript, together with the fMOST dataset as a 4th bar plot next to the Patch-clamp and the two Patch-Seq comparisons (as in the bar plot above on page 1) without normalization to CAJAL as 100%. Separate plot for each used dataset would show the ground-truth of the performance of CAJAL.

We have taken the reviewer's suggestion into consideration and made changes to the revised manuscript accordingly. Specifically, we have replaced Figure 2d with a bar plot that displays the accuracy, CTS score, and Matthews correlation coefficient of each method in each of the four datasets. These metrics are no longer normalized with respect to CAJAL, providing a straight representation of the performance of each method.

Such an interpretation would show the same data, but it would also draw the attention of the readers that Accuracy strongly depends on the chosen dataset. As the authors found the fMOST dataset, where the CAJAL performed well, it indeed supported the usefulness of the CAJAL within that dataset. At the same time, we could observe in the originally used Patch-Seq datasets that CAJAL did not outperform existing packages. The authors should explain what the characteristic of a dataset is (brain region, cell types or a reconstruction method, other technical factor...etc) what makes the CAJAL outperform other packages, this would explain the advantage of the package and guide the potential future users to specifically apply it on a certain type of dataset.

While it is not possible to draw conclusions about the impact of brain regions, cell types, and reconstruction methods based on four datasets, we did observe some differences in the performance of the various methods. These differences may be attributed to a range of factors, including differences in the modeling assumptions, the use of predefined morphological features, and the ability to produce a morphological distance function between CAJAL and other methods for neuronal morphometry. In the revised manuscript, we have summarized the key factors in Supplementary Figure 2, presenting them in the form of a flowchart, as suggested by the editor. We believe that this chart will be of utility to users, providing them with guidance on when to best utilize CAJAL in their research.

High performance is one of the major statements of the manuscript. The authors explained that they normalized the values of accuracy to CAJAL's performance. The Accuracy by itself is a percentage, expressing that out of 100 neurons in case of their example, accuracy of SNT would be ~23 cells and accuracy of CAJAL would be ~29 cells. This simply means that CAJAL would score 29 instead of 23, which may be convincing to use the package, as it is truly an improvement. Normalization shows 1.28, which means 28% improvement, which is deceiving in this form. Without this normalization of percentages, the performance of CAJAL does look better in certain cases but does not look outstanding on the accuracy plots I suggested, which are presenting ground-truth without exaggeration.

In the manuscript we have provided comprehensive information on the accuracy of each method (Figure 2d and Supplementary Table 2), including the non-normalized quantification of accuracy. Following the reviewer's suggestion, we do not present relative accuracies in the main figure.

This being said, I appreciate the effort of the authors to improve the interpretation of the performance of CAJAL. The authors proved that the CAJAL works per se, and on certain datasets could outperform some existing packages by a given percentage. At the same time, it is clear that the CAJAL is only superior to some existing packages on certain datasets, and for some reason not when used on Patch-Seq anatomy.

References

- 1 Pincus, Z. & Theriot, J. A. Comparison of quantitative methods for cell-shape analysis. *J Microsc* **227**, 140-156, doi:10.1111/j.1365-2818.2007.01799.x (2007).